# Neoadjuvant therapy with immune checkpoint blockade, antiangiogenesis, and chemotherapy for locally advanced gastric cancer

Song Li [1,9], Wenbin Yu [2,9], Fei Xie [3], Haitao Luo [4], Zhimin Liu [5], Weiwei Lv [6], Duanbo Shi [7], Dexin Yu [6], Peng Gao [7], Cheng Chen [2], Meng Wei [2], Wenhao Zhou [4], Jiaqian Wang [4], Zhikun Zhao [4], Xin Dai [8], Qian Xu [1], Xue Zhang [1], Miao Huang [1], Kai Huang [1], Jian Wang [1], Jisheng Li [1], Lei Sheng [2] & Lian Liu [1] ✉

Despite neoadjuvant/conversion chemotherapy, the prognosis of cT4a/bN+ gastric cancer is poor. Immune checkpoint inhibitors (ICIs) and antiangiogenic agents have shown activity in late-stage gastric cancer, but their efficacy in the neoadjuvant/conversion setting is unclear. In this single-armed, phase II, exploratory trial (NCT03878472), we evaluate the efficacy of a combination of ICI (camrelizumab), antiangiogenesis (apatinib), and chemotherapy (S-1 ± oxaliplatin) for neoadjuvant/conversion treatment of cT4a/bN+ gastric cancer. The primary endpoints are pathological responses and their potential biomarkers. Secondary endpoints include safety, objective response, progression-free survival, and overall survival. Complete and major pathological response rates are 15.8% and 26.3%. Pathological responses correlate significantly with microsatellite instability status, PD-L1 expression, and tumor mutational burden. In addition, multi-omics examination reveals several putative biomarkers for pathological responses, including *RREB1* and *SSPO* mutation, immune-related signatures, and a peripheral T cell expansion score. Multi-omics also demonstrates dynamic changes in dominant tumor subclones, immune microenvironments, and T cell receptor repertoires during neoadjuvant immunotherapy. The toxicity and post-surgery complications are limited. These data support further validation of ICI- and antiangiogenesis-based neoadjuvant/conversion therapy in large randomized trials and provide candidate biomarkers.

Gastric cancer remains a major killer globally, ranking fifth for incidence and fourth for mortality[1]. Radical resection is the best treatment option for curing and prolonging survival[2]. In the previous phase III MAGIC[3] and FFCD[4] clinical trials, perioperative chemotherapy showed higher 5-year survival rates than surgery alone. The FLOT regimens further elevated R0 resection rates and prolonged overall survival (OS) compared to regimens without paclitaxel[5]. However, the cT4 patients accounted for fewer than 10% in that trial[5]. In the recent PRODIGY study[6], neoadjuvant DOS regimen (docetaxel, oxaliplatin, and S-1[Tegafur/Gimestat/Oxonate]) and adjuvant S-1 significantly

improved the 3-year progression-free survival (PFS) rate versus post-operative S-1 (66.3 *vs.* 59.8 months) and resulted in a 10.4% pathological complete response. Although 70% of the patients were cT4 in this trial, cT4a accounted for up to 89%. In addition, the cT4N+ group showed an inferior outcome to cT4N- and cT2-3N+ patients, and there was no subgroup analysis solely for cT4bN+ patients[6].

Recently, immune checkpoint inhibitors (ICIs) achieved superior outcomes to placebo in the third-line treatment for advanced gastric cancer in phase III randomized controlled trials[7]. In the first-line setting, combined ICI and chemotherapy showed prolonged OS and PFS compared to chemotherapy alone and reduced the risk of death by 20–35% in patients with PD-L1 CPS ≥ 5 in CheckMate 649[8] and ORIENT-16[9]. Theoretically, the neoadjuvant setting is optimal for immunotherapy due to intact immune systems, ample neoantigens, and low tumor clonalities[10]. Investigation of neoadjuvant ICI-based therapy has succeeded in resectable non-small cell lung cancer[11,12]. Whether it works in gastric cancer, particularly in locally advanced gastric cancer, remains incompletely explored.

Tumor angiogenesis plays an essential role in tumor progression. Like ICIs, antiangiogenic agents target tumor microenvironment (TME) components other than tumor cells and synergize with ICIs by promoting CD8+ T lymphocyte infiltration and activation[13]. Ramucirumab, an anti-VEGFR2 antibody[14], and apatinib, a VEGFR2 tyrosine kinase inhibitor[15], have been shown to prolong OS and were approved for second-line and third-line treatment of advanced gastric cancer, respectively. They have been shown to reprogram TME, reverse immune-suppressive to inflamed state, and enhance the efficacy of ICIs in several phases I/II studies[16–18]. Therefore, adding antiangiogenic agents to ICIs plus chemotherapy regimens may enhance neoadjuvant efficacy.

In this phase II trial, we explore the efficacy and safety of a neoadjuvant/conventional combination therapy with anti-PD1 antibody (camrelizumab), antiangiogenic agent (apatinib), and chemotherapy (S-1 ± Oxaliplatin) in stage T4a/bN+M0 gastric cancer patients. Complete and major pathological response rates are 15.8% and 26.3%. Sequential multi-omics tests, including whole-exome sequencing (WES), transcriptome sequencing, and T cell receptor (TCR) sequencing, reveal several putative biomarkers for pathological responses and dynamic changes in dominant tumor subclones, immune microenvironments, and T cell receptor repertoires during neoadjuvant immunotherapy.

## Results

### Patient characteristics

Between May 2019 and August 2021, 25 patients were enrolled, between 48 to 70 years old, 19 male, 11 cT4aN+ and 14 cT4bN+ (Table 1). All patients completed neoadjuvant therapy and re-evaluation for surgery (Fig. 1b). The detailed treatment regimens and cycle numbers are listed in Supplementary Table 1. Among them, 4 were unresectable and 2 refused surgery, leaving 19 patients who received resection and were evaluable for pathological response (Supplementary Fig. 1). The median follow-up was 24.7 months (Quartiles, 20.9–31.8 months; Fig. 1b).

### Clinical efficacy

In 11 patients with cT4aN+ gastric cancer, 9 (81.8%) achieved radiological downstaging. Three (27.3%) patients had partial responses, and 8 (72.7%) had stable diseases (Fig. 1c). Nine of them underwent surgery with R0 resection. Two resections were considered palliative due to peritoneal metastasis found during surgery (Fig. 1b). At a median of 26.7 (Quartiles, 25.0–36.4) months of follow-up, 7 of 9 (77.8%) patients who had undergone radical resection were alive, and 5 (55.6%) were recurrence-free (Fig. 1d).

In 14 patients with cT4bN+ patients, 10 (71.4%) achieved radiological downstaging. Four (28.6%) patients had partial responses, and

**Table 1 | Baseline characteristics of patients according to clinical T staging**

| Characteristic | All patients (n = 25) | cT4a (n = 11) | cT4b (n = 14) |
|---|---|---|---|
| Age, median (range) | 63 (48–70) | 64 (51–69) | 61 (48–70) |
| Gender, n (%) | | | |
| Female | 6 (24) | 2 (18) | 4 (29) |
| Male | 19 (76) | 9 (82) | 10 (71) |
| ECOG, n (%) | | | |
| 0 | 14 (56) | 7 (64) | 7 (50) |
| 1 | 11 (44) | 4 (36) | 7 (50) |
| Clinical T staging, n (%) | | | |
| T4a | 11 (44) | 11 (100) | 0 (0) |
| T4b | 14 (56) | 0 (0) | 14 (100) |
| Clinical N staging, n (%) | | | |
| N2 | 3 (12) | 1 (9) | 2 (14) |
| N3 | 22 (88) | 10 (91) | 12 (86) |
| PD-L1 (CPS), n (%) | | | |
| ≥10 | 1 (4) | 1 (9) | 0 (0) |
| 5–10 | 1 (4) | 0 (0) | 1 (7) |
| 1–5 | 1 (4) | 1 (9) | 0 (0) |
| <1 | 21 (84) | 9 (82) | 12 (86) |
| Unknown | 1 (4) | 0 | 1 (7) |
| MMR/MSI, n (%) | | | |
| dMMR/MSI-H | 4 (16) | 2 (18) | 2 (14) |
| pMMR/MSS | 21 (84) | 9 (82) | 12 (86) |
| ERBB2 amplification, n (%) | | | |
| YES | 4 (16) | 1 (9) | 3 (21) |
| No | 21 (84) | 10 (91) | 11 (79) |
| Signet cells, n (%) | | | |
| Yes | 9 (36) | 7 (64) | 2 (14) |
| No | 16 (64) | 4 (36) | 12 (86) |
| Lauren's classification, n (%) | | | |
| Diffused | 16 (64) | 7 (64) | 9 (64) |
| Intestinal/mixed | 9 (36) | 4 (36) | 5 (36) |

*ECOG* Eastern Cooperative Oncology Group, *MMR* DNA mismatch repair, *MSI* microsatellite instability, *dMMR* deficient MMR, *pMMR* proficient MMR, *MSI-H* MSI high, *MSS* microsatellite stable.

10 (71.4%) patients had stable diseases (Fig. 1c). In 12 per-protocol patients, 8 (72.7%) underwent radical resection, 1 found peritoneal metastasis during surgery, and 3 were not resectable by surgeons' evaluation (Fig. 1b). At a median of 23.7 (Quartiles, 21.1–31.6) months of follow-up, 4 of 8 (50.0%) patients who had undergone radical resection were recurrence-free, and 5 (62.5%) of them were alive (Fig. 1d).

### Pathological findings

In total, 19 patients were evaluable for pathological response. Baseline frequencies of microsatellite instability-high (MSI-H), PD-L1 positive (CPS ≥ 1), *ERBB2* amplification, and Lauren's diffuse type were 15.8% (3/19), 16.7% (3/18), 21.1% (4/19), and 52.6% (10/19; Fig. 2a). The median pathological regression was 40% (Quartiles, 7–93%; Fig. 2a).

Among them, 3 (15.8%, 95% CI 3.4–39.6%) achieved complete pathological response (CPR), 5 (26.3%, 95% CI 9.1–51.2%) major pathological response (MPR) and 8 (42.1%, 95% CI 20.0–66.5%) partial pathological response+ (PPR+; Fig. 2a and Table 2). According to the Becker regression criteria, 3 (15.8%, 95% CI 3.4–39.6%) achieved TRG1a and 2 (10.5%, 95% CI 1.3–33.1%) achieved TRG1b (Table 2). The combined TRG1a/b rate was 26.3% (95% CI 9.1–51.2%). Morphological changes in post-treatment samples with MPR included extracellular

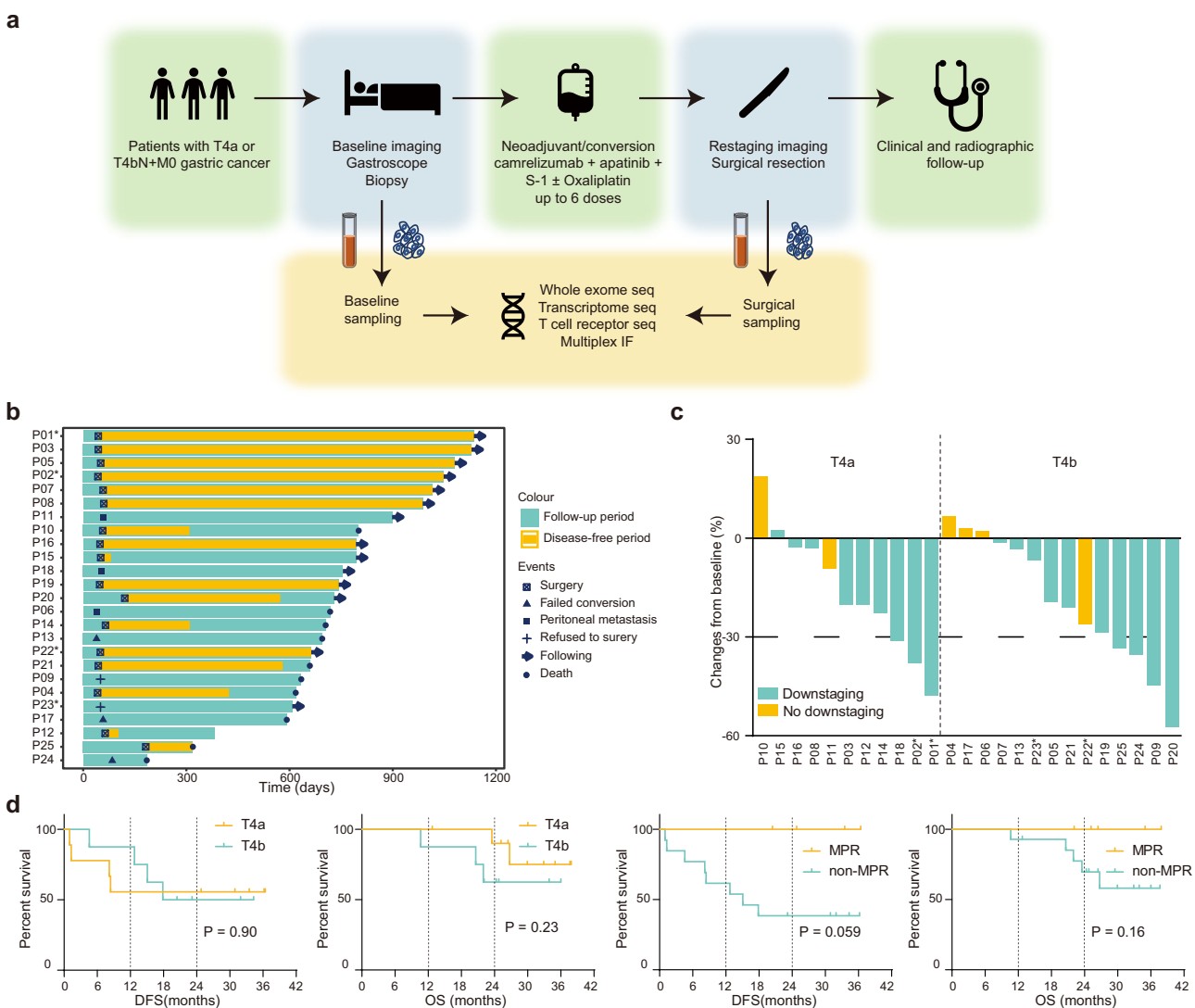

**Fig. 1 | Study design and clinical efficacy. a** Trial schema of the study. Patients with cT4N+ received neoadjuvant camrelizumab, apatinib, S-1 with or without oxaliplatin, followed by surgical resection. The primary endpoints were pathological responses and their potential biomarkers. Tumor and blood samples were collected at baseline and at the time of surgery for multi-omics analysis. **b** Swimmer plot of 25 patients involved in this trial. **c** Tumor size changes from baseline according to radiological imaging. **d** Kaplan-Meier curves of disease-free survival and overall survival stratified by clinical T stages (left two panels) and pathological responses (right two panels). Log-rank test was used. * represents patients with microsatellite instability-high tumors in **b** and **c**. Source data are provided as a Source Data file.

mucin pools, fibrosis, and lymphocyte infiltration (Fig. 2d). Pathological responses overlapped with radiological responses well, with 4/5 (80%) patients with MPR having radiological partial responses (Fig. 2b). CT scan images in Fig. 2e show partial responses of two patients with MPR after neoadjuvant therapy (P1 and P18).

Among 5 patients with MPR, 2 were MSI-H and PD-L1 positive, 1 was MSI-H and PD-L1 negative, 1 was microsatellite stable (MSS) and PD-L1 positive with *ERBB2* amplification, and 1 was MSS and PD-L1 negative at baseline (Fig. 2a). By contrast, non-MPR patients were all MSS and baseline PD-L1 negative. After neoadjuvant therapy, 2 PD-L1 negative MPR (100%) and 4 non-MPR (30.8%) tumors turned into PD-L1 positive (Fig. 2c).

We further investigated an MPR case with MSS and PD-L1 negative (P18) and two non-MPR cases (P10 and P20) by multiplex immunofluorescence (Fig. 2f **and** Supplementary Fig. 2). In the pre-treatment specimen of P18, tumor cells were densely distributed, with rare PD-L1 expression (Fig. 2f). Most CD8+ T cells were PD-1 positive. Regulatory T ($T_{reg}$) cells and macrophages were sporadic. After treatment, tumor cells got degenerated and loose in nest-like structures. An influx of

immune cells, especially CD8+ cells, was observed around the mucin pool. Many of the stromal cells were PD-L1 positive, while the tumor cells were still PD-L1 negative. Besides, some macrophages and tumor cells were close to each other. In the two patients with non-MPR (P10 and P20), there was also CD8+ T cell infiltration after treatment, yet to a lesser degree than in P18 (Supplementary Fig. 2). In P20, macrophages were increased after treatment and $T_{reg}$ cells were found enriched in both pre- and post-treatment specimens (Supplementary Fig. 2). WES and TCR sequencing showed drops in tumor mutational burden (TMB) and tumor neoantigen burden (TNB) (Fig. 2g–h) and expansion of hyperexpanded T cell clones (Fig. 2i) in post-treatment samples with MPR.

**Feasibility and safety**

All patients received a mean of 2.6 cycles of neoadjuvant treatment. Adverse events occurred in all patients, but only two events were of grade 3 or higher (neutropenia; Supplementary Fig. 3). The most frequent adverse events were nausea (56%), anorexia (48%), fatigue (40%), neutropenia (36%), hypothyroidism (20%), thyroiditis (20%),

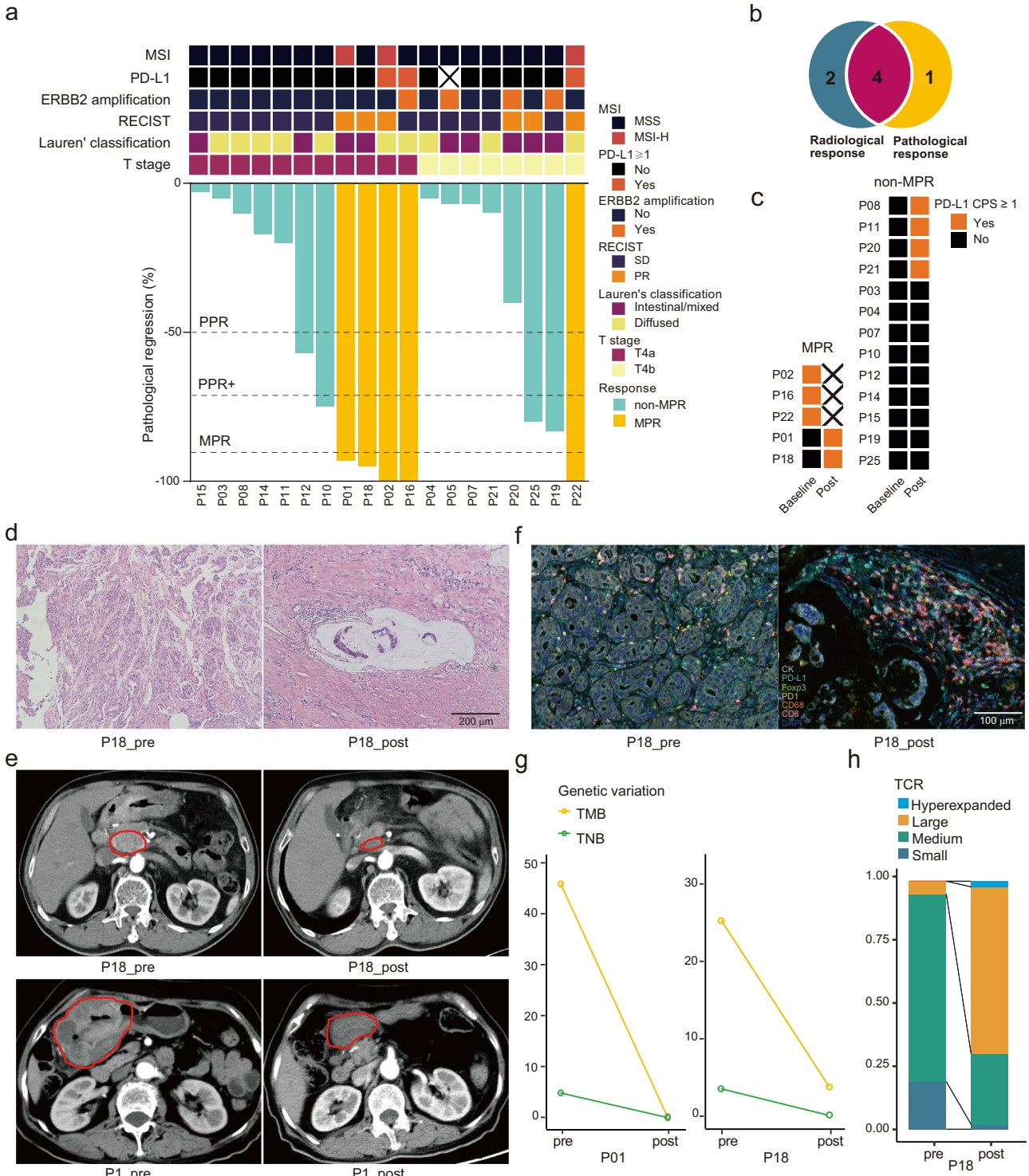

**Fig. 2 | Pathological findings of resected tumors. a** Pathological responses in T4a ($n = 11$) and T4b ($n = 8$) gastric cancer patients. Baseline features are shown in the top panel, while percentages of pathological regression are shown at the bottom. Dot lines indicate cutoff values of partial pathological response (PPR), PPR+, and major pathological response (MPR). **b** Overlap of partial response by radiological examination and MPR. **c** Dynamic changes of PD-L1 expression positivity. **d** Representative hematoxylin and eosin staining sections of tumor specimens obtained from patient P18, who obtained MPR. **e** Representative computed tomographic imaging of a lymph node (top) and the stomach wall (bottom) of two patients who received MPR. **f** Multiplex immunofluorescence staining of patient P18 who received MPR. Visible structures include cytokeratin-positive tumor cells (white), PD-L1+ cells (cyan), FoxP3+ regulatory T cells (green), PD-1+ cells (yellow), CD68+ macrophages (orange), CD8+ T cells (magenta), and nuclei (blue). Staining was performed only once in **d** and **f**. **g** Changes in tumor mutational burden (TMB) and tumor neoantigen burden (TNB) in patients P01 and P18 with MPR. **h** Changes in peripheral TCR clones in patient P18. Source data are provided as a Source Data file.

**Table 2 | Efficacies of the neoadjuvant/conversion treatment**

|                            | Total (n = 25)   | cT4aN+ (n = 11) | cT4bN+ (n = 14) |
|----------------------------|------------------|-----------------|-----------------|
| Tumor down-staging rate    | 76.0% (19/25)    | 81.8% (9/11)    | 71.4% (10/14)   |
| Objective response rate    | 28.0% (7/25)     | 27.3% (3/11)    | 28.6% (4/14)    |
| R0 resection rate          | 82.6% (19/23)    | 100.0% (11/11)  | 72.7% (8/12)    |
| CPR rate                   | 15.8% (3/19)     | 18.2% (2/11)    | 12.5% (1/8)     |
| MPR rate                   | 26.3% (5/19)     | 36.4% (4/11)    | 12.5% (1/8)     |
| PPR + rate                 | 42.1% (8/19)     | 45.5% (5/11)    | 37.5% (3/8)     |
| TRG 1a                     | 15.8% (3/19)     | 18.2% (2/11)    | 12.5% (1/8)     |
| TRG 1b                     | 10.5% (2/19)     | 18.2% (2/11)    | 0 (0/8)         |
| TRG 1a/b                   | 26.3% (5/19)     | 36.4% (4/11)    | 12.5% (1/8)     |
| TRG 2                      | 21.1% (4/19)     | 18.2% (2/11)    | 25.0% (2/8)     |
| TRG 3                      | 52.6% (10/19)    | 45.5% (5/11)    | 62.5% (5/8)     |

TRG grades were accessed according to the Becker TRG system. *CPR* complete pathological response, *MPR* major pathological response, *PPR* partial pathological response, *TRG* tumor regression grading.

infusion reaction (16%), and headache (16%; Supplementary Fig. 3). There were no previously unreported toxic effects. No discontinuation of neoadjuvant treatment occurred due to toxicity.

Among 20 patients who underwent surgery, the median interval between the last dose of apatinib and surgery was 16 days (Quartiles, 14–19 days; Fig. 1b). One surgical delay (93 days) occurred due to treatment-related pneumonia, and complete resection was performed with a pathological result PPR+ after remission of pneumonia.

Seven patients (35%) experienced grade 1 or 2 postoperative complications, and there were no grade 3 or higher complications (Supplementary Fig. 4). The most common complications were pleural effusion (15%), respiratory tract infection (10%), and atelectasis (10%). Anastomotic leaks occurred in one patient (5%). The median duration of hospital stay was 12.0 (Quartiles, 10.5–13.0) days. There was no re-operation nor death within 30 days.

### Links between gene mutations and efficacy of neoadjuvant therapy

Consistent with the TCGA STAD cohort[19], the most frequent mutations occurred on *TTN*, *TP53*, *SPTA1*, etc. (Fig. 3a and Supplementary Data 1). Some mutations co-existed before and after neoadjuvant therapy in the same patient and may contribute to resistance to therapy (Fig. 3a). At baseline, TMB and TNB were significantly higher in MPR than in non-MPR tumors (Fig. 3a, b). Because of the association between MSI-H and high mutational burden, we excluded MSI-H patients for further analysis and found TMB and TNB were higher in MPR than in non-MPR tumors, yet without statistical significance (Fig. 3c). TMB decreased significantly in the entire population after treatments, and also when MSI-H or MPR patients were excluded (Fig. 3d). Considering the possible association between TMB and purity, we performed a simulation to balance purities between pre- and post-treatment samples. After the simulation, there were still significant decreases in purity-adjusted TMB in these patients (Fig. 3e). To better understand mutational selection during the evolution under treatment, we used a package "dNdScv" to estimate the relative rates of nonsynonymous and synonymous mutations (dN/dS)[20]. We found that dN/dS significantly dropped after treatment in PPR+ patients, suggesting a negative selection and reduced subclone diversity (Supplementary Fig. 5b).

*RREB1* and *SSPO* mutations were found in 80% (4/5) and 80% (4/5) of the baseline samples with MPR, but in 0 and 7% (1/14) of non-MPR specimens, respectively, possibly predicting good responses as putative biomarkers (Fig. 3a). *SSPO* is a pseudogene in humans with unknown roles in cancer[21]. Patients with *SSPO* mutation showed significantly prolonged survival in the TCGA STAD cohort but not in the pan-cancer cohort (Fig. 3f). Several other genes, including ADAMTS12 (5/5), KIAA1549 (4/5), PITX2 (4/5), RECQL4 (4/5), and TRPS1 (4/5) also frequently mutated in MPR patients, but rarely in non-MPR patients (Supplementary Data 2).

Based on the changes of purity-corrected VAF from pre- to post-treatment tumors, nonsynonymous mutations were divided into four subtypes, "gain" (occurred only in post-treatment samples), "lost" (occurred only in pre-treatment samples), "increase" (VAF increased after treatment), and "decrease" (VAF decreased after treatment; Fig. 3g and Supplementary Fig. 6). Pathological regressions were significantly correlated with proportions of "lost" mutations (Fig. 3h).

### Clonal evolution during neoadjuvant therapy and drug-resistance

Therapeutic interventions can destroy sensitive cancer clones but provide a selective pressure for resistant variant expansion[22]. We used PyClone to study the effects of neoadjuvant therapy on tumor clonal evolution (Fig. 3i and Supplementary Fig. 7). Subclone contractions were observed in many patients after the neoadjuvant treatment (Fig. 3i). Expansions were observed in some patients, such as P03, P04, and P07 (Fig. 3i).

### Differential gene expression and TME in patients with different responses

Next, the transcriptome data and their association with responses were analyzed. We identified 209 differential expression genes (DEGs) between baseline samples of patients with MPR and non-MPR. These DEGs were mainly enriched in metabolisms of vitamin and fat, cell to cell/matrix adhesion, and multiple signaling pathways, including PI3K-Akt, MAPK, Ras, and Wnt pathways (Fig. 4a, b). Then, we explored the DEGs between pre- and post-treatment samples (Supplementary Fig. 8a, b). The neoadjuvant therapy impacted IL-17 and TNF pathways in both MPR and non-MPR patients (Supplementary Fig. 8c, d). Many differential pathways between baseline MPR and non-MPR samples, including metabolisms of vitamin and fat, cell to cell/matrix adhesion, and multiple (PI3K-Akt, MAPK, Ras, Wnt, etc.) signaling pathways, were altered by the treatment in MPR patients (Supplementary Fig. 8c), but not in non-MPR patients (Supplementary Fig. 8d).

At baseline levels, there were significantly higher levels of immune checkpoint genes (*e.g.*, CD274 and CTLA4) and cytolytic genes (*e.g.*, GZMB, NKG7, and PRF1) in PPR+ samples than in non-PPR+ samples (Fig. 4c, d and Supplementary Fig. 9a). In patients with PPR+, the neoadjuvant therapy suppressed immune-suppressive genes, such as CD274, FOXP3, and IDO1, and upregulated cytolytic genes, including CD8A and GZMH, but these differences have no statistical significance (Fig. 4e and Supplementary Fig. 10a). When MSI-H patients were excluded, we found similar trends (Supplementary Figs. 9b and 10b). The decreased *CD274* mRNA seems to differ from the IHC data (Fig. 2c), probably due to post-translational regulation of its expression or due to poor overlap between samples in these two experiments. In addition, some cytolytic genes (*e.g.*, GZMH, NKG7, and GZMA) were significantly elevated in the post-treatment specimens in non-PPR+ patients (Supplementary Fig. 10c).

Immune-related signatures were also analyzed (Fig. 4f). Cytolytic, IFN-gamma, and T-cell exhaustion were significantly higher in PPR+ patients than in non-PPR+ patients at baseline levels (Fig. 4g and Supplementary Fig. 11). Neoadjuvant therapy promoted cytolytic and CD8+ effector signatures in both PPR+ and non-PPR+ patients, yet with no significance (Fig. 4h and Supplementary Fig. 12).

xCell was used to estimate 23 immune cell types in TME (Fig. 5a and Supplementary Fig. 13). In PPR+ tumors, proportions of DC, CD8+ T cells, T helper cells, and M1 macrophages were augmented after treatment, while $T_{reg}$ cells decreased (Fig. 5c). In non-PPR+ tumors, the changes of DC, CD8+ T cells, and M1 macrophages were similar, but only 2/9 patients showed reduced $T_{reg}$ cells and 4 showed upregulated

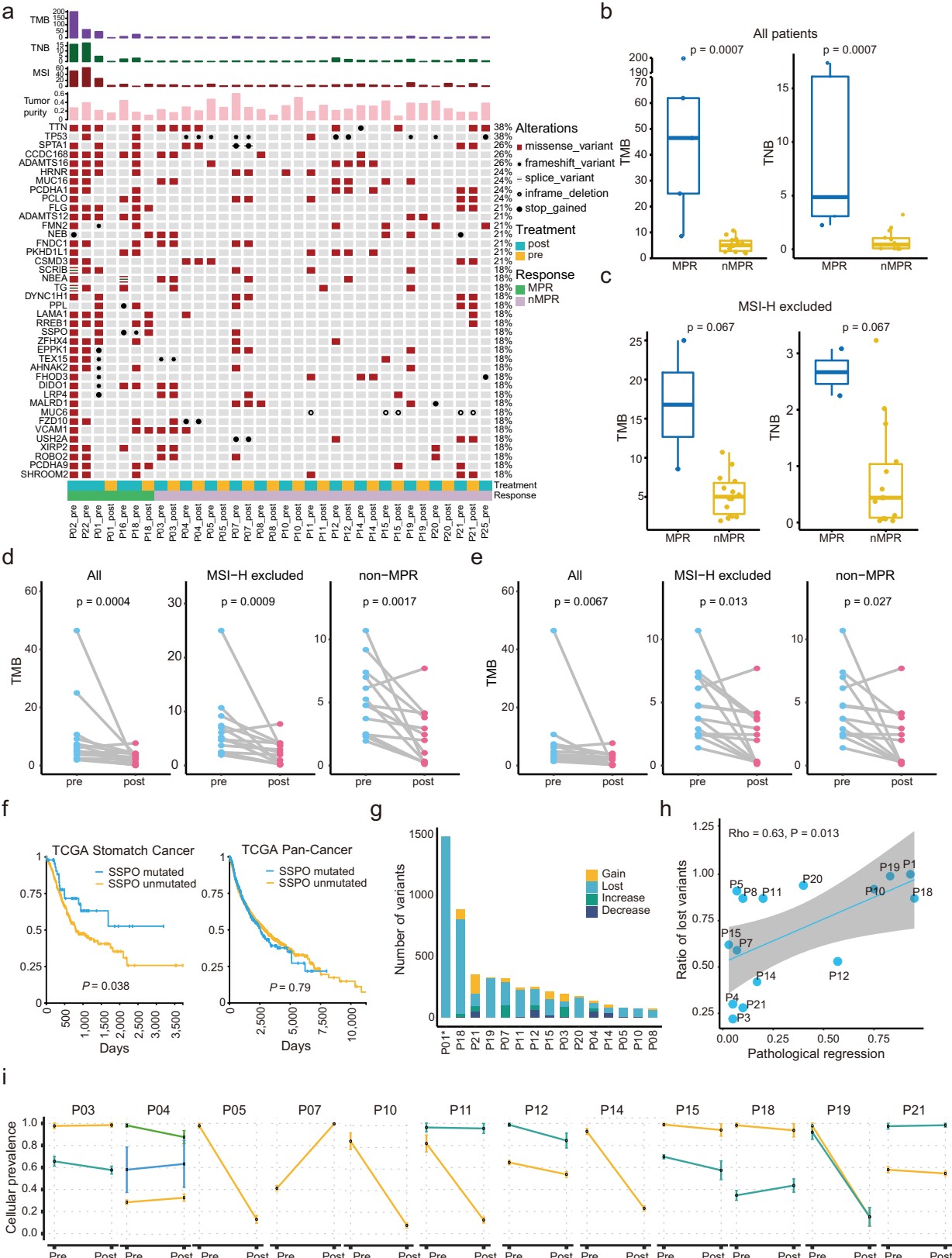

$T_{reg}$ cells (Fig. 5d). The Cibersort software confirmed the dynamic changes of the above cells and provided information about the cell subtypes (Fig. 5b). For example, polarized macrophages (M1 and M2) significantly increased, while the naïve cells (M0) decreased, reflecting a lineage differentiation from M0 to M1/2 during treatment (Fig. 5b).

TME subtypes were accessed by a KNN model trained by previous data (Fig. 5a)[23]. All PPR+ and 5/9 non-PPR+ (P03, P04, P08, P15, and P20) tumors shifted from "depleted" or "fibrotic" to "immune" or "immune/fibrotic" types during the treatment (Fig. 5e). We also used Immunophenoscore to compare the evolution of local immune status at four dimensions, including MHC molecules (MHC),

**Fig. 3 | Genomic characteristics and clonal evolution following neoadjuvant treatment. a** Mutational landscape of pre-/post-treatment samples in 19 patients. The values of TMB, TNB, MSI scores, and tumor purity are shown in the upper panel. **b** Differences in TMB and TNB between MPR (*n* = 5) and non-MPR patients (*n* = 14). **c** Differences in TMB and TNB between MPR (*n* = 2) and non-MPR (*n* = 14) patients when MSI-H patients were excluded. Centers, boxes, and whiskers indicate medians, quantiles, and minima/maxima, respectively, in **b** and **c**, and two-sided Wilcoxon rank-sum test was used for comparison. **d-e** TMB changes in all paired specimens (*n* = 15), MSI-H-excluded patients (*n* = 14), and non-MPR patients (*n* = 13) before (**d**) and after (**e**) purity adjustment. Two-sided Wilcoxon signed-rank test was used for comparison in **d** and **e**. **f** Overall survival curves with and without *SSPO*

mutation in TCGA STAD (n = 433) and Pan-Cancer (*n* = 9034) cohorts. Log-rank test was used for curve comparison. **g** Frequencies of mutation types that are classified according to VAF changes. "Lost" and "Gain" indicate unique mutations in pre- and post-therapy samples, respectively. "Increase" and "Decrease" indicate mutations whose VAF increased and decreased in post-therapy samples, respectively. **h** Correlation between "Lost" variants ratios and percentage of pathological regression across different patients, accessed by Spearman's rank correlation coefficient (rho). The regression line is blue, and the shading indicates the 95% confidence interval. **i** Changes in cellular prevalence of tumor subclones. Error bars indicate standard deviation. Source data are provided as a Source Data file.

immunomodulators (CP), effector cells (EC), and suppressor cells (SC; Fig. 5f and Supplementary Fig. 14). After the neoadjuvant therapy, EC and MHC scores increased in all PPR+ patients and some non-PPR+ patients (e.g., P04, P08, P20, P21) (Fig. 5f and Supplementary Fig. 14). However, SC and CP significantly increased in these non-PPR+ patients but remained stable in PPR+ patients (exemplified in Fig. 5f). The other non-PPR+ tumors (e.g., P07, P08, P11, P12, and P15) had no obvious changes after the treatment.

### Dynamics of T cell clones and TCR repertoire during treatment

By TCR sequencing, we analyzed the TCR repertoire of peripheral T cells from these patients. The landscape of TCR repertoire shows that samples from the same patients shared the most TCR sequences, while only small portions of TCR were shared among different patients (Fig. 6a and Supplementary Fig. 15). Scores of richness and evenness were used to quantify TCR diversity, but no significant differences were found between PPR+ and non-PPR+ or between pre- and post-treatment samples (Supplementary Fig. 16 and Supplementary Table 2).

Then, we compared the dynamic changes of TCR clonality in 4 PPR+ and 6 non-PPR+ patients (Fig. 6b). A clonal expansion (CE) score was calculated based on the frequency change of top 20 T cell clones (CE score = *frequency of top 20 clones in post-treatment samples – that in pre-treatment samples*). The CE scores correlated significantly with pathological regressions; they were all positive in four PPR+ patients and negative in six non-PPR+ patients (Fig. 6c). This suggests a remarkable expansion of T cell clones in the responders, and the CE score could be an excellent biomarker for pathological response.

We also divided the T cell clones in each sample into four categories according to the TCR frequencies, small (≤0.0001), medium (0.001–0.0001), large (0.01–0.001), and hyperexpanded (>0.01) (Fig. 6d). In PPR+ patients, the frequencies of hyperexpanded clones were significantly elevated after treatment, while those of small clones were reduced (Fig. 6e). These differences were not observed in non-PPR+ patients. These results further indicate that patients with T-cell clone expansion have better pathological responses than others.

In addition, we investigated TCR V and J segment usages (Fig. 6f, g). Different degrees of V and J usage changes were observed during treatment. Remarkable expansions of *TRBV20-1* (Fig. 6f) and *TRBJ2-5* (Fig. 6g) co-occurred in both MPR patients (P16 and P18) but not in other patients, suggesting that TRBV20-1 and TRBJ2-5 might contribute to anti-tumor immunity.

### Discussion

For locally advanced gastric cancer, 5-year survival rates are only 30.5%, 20.1%, and 8.3% for IIIA, IIIB, and IIIC patients[2]. Neoadjuvant chemotherapy has been widely used to improve R0 resection rates and DFS[24], but its efficacy is still limited by low pathological regressions[25]. By combining ICI, antiangiogenic agents, and chemotherapy in the neoadjuvant/conversion setting, we achieved 15.8% CPR, 26.3% MPR, and satisfying safety and feasibility. By multi-omics technique, we investigated indicators associated with pathological responses and

evolutions of tumors, immune TME, and T cell clones during neoadjuvant immunotherapy.

Based on preliminary data, the neoadjuvant ICI-based therapy led to good outcomes in pathological responses, especially in MSI-H or PD-L1 positive patients. This was consistent with several recent phase I/II single-armed studies in the 2021 ASCO annual meeting, which used ICI plus chemotherapy to treat cT3-4 or N+ gastric cancer patients and achieved >90% R0 resection rates, 0–25% CPR, and 22–42% MPR[26–31]. Compared with these studies, our trial recruited patients with more advanced cancer, all being cT4N+, 56% initially unresectable cT4bN+, and 64% Lauren's diffused type patients. Therefore, patients in our study may have inferior outcomes to those in the above trials; cT4N+ patients faced nearly twice the risks of recurrences as cT4N0 or cT2–3N+ patients (40.1% *vs.* 25.0% and 22.2%)[6]. Neoadjuvant chemotherapy by FLOT4 also achieved a similar CPR rate, but cT4 patients accounted for only 9% in this trial[32]. Meanwhile, while CPR hardly occurred (3%) in the diffused type in the FLOT4 trial, CPR was 30% in this pathological type in our study[32]. In addition, our cT4bN+ patients with conversion therapy received over 70% downstaging and radical resection and 12.5% MPR rates. Unlike the previous report on lung cancer[11], we observed consistent responses in radiology and pathology. Notably, patients with MSI-H tumors showed a 100% (3/3) MPR rate in our study. Even though MSI-H is a good predictor of responses, patients with advanced MSI-H gastric cancer received only 47–57% ORR from ICI monotherapy[33] and 55% ORR from ICI plus chemotherapy in CheckMate 649[34].

As another difference from other neoadjuvant ICI-based trials using ICI plus chemotherapy, we added antiangiogenic agents for combination. Blockage of VEGF/VEGFR has been reported to inhibit angiogenesis and immune suppression in TME, synergizing with ICI to promote local immune responses[13,35]. Another phase II trial combining ICI and concurrent chemoradiotherapy achieved excellent rates of R0 resection (95.0%), MPR (73.7%), and CPR (42.1%) in localized advanced gastric cancer, 17.9% of which were T4bN+, with no molecular pathology reported[36]. This study and ours indicate that adding more treatment methods to ICI plus chemotherapy might reprogram TME to be "hotter" and improve efficacy. On the other hand, these data are from small trials. Introducing more treatments, especially radiotherapy, may lead to higher risks of toxicity, so a large randomized controlled trial is needed to explore a synergistic combination regimen to optimize effectiveness, feasibility for surgery, and tolerated toxicity.

Peri-surgical antiangiogenesis may be associated with safety concerns because it is an essential step in wound healing[37]. In former trials, neoadjuvant bevacizumab increased incidences of postoperative anastomotic leak and wound healing complications after oesophagogastrectomy[38]. Ramucirumab plus FLOT also had higher surgical morbidity than FLOT (44% *vs.* 37%)[39]. By contrast, our preliminary data showed that apatinib did not show high-incidence morbidities, including anastomotic leakage (5%) and wound healing complications (5%). The reason might lie in that apatinib is a small molecular tyrosine kinase, which has a much shorter half-life (about 9 h) than antibodies (*e.g.*, about 20 days for bevacizumab)[40,41].

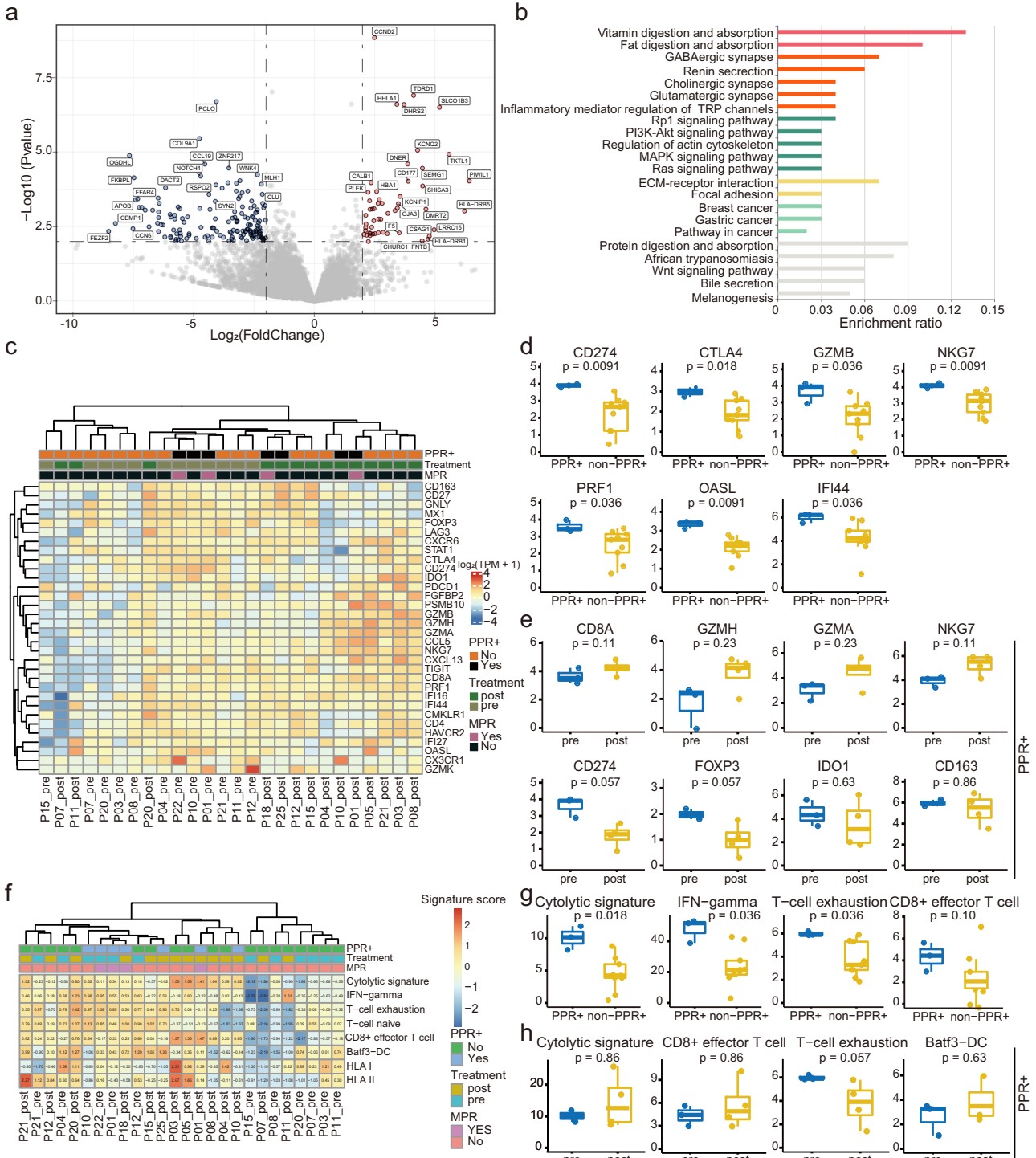

**Fig. 4 | Transcriptomic features in patients with different responses. a** Volcano plot showing differentially expressed genes between baseline MPR ($n = 2$) vs. non-MPR ($n = 10$) group. Color dots denote genes that passed the p-value and fold change thresholds. **b** Functional pathway enrichment by differential expressed genes at baseline. Similar pathways are clustered and stained with the same colors. **c** Heatmap for immune-related gene expression across samples in 15 patients. **d** Baseline levels of immune-related genes that were altered between PPR+ ($n = 3$) and non-PPR+ (n = 9) groups. **e** Level changes of immune-related genes between

pre- ($n = 3$) and post-treatment ($n = 4$) PPR+ patients. **f** Heatmap for immune-related signatures across samples in 15 patients. **g** Baseline levels of immune-related signatures that were altered between PPR+ ($n = 3$) and non-PPR+ ($n = 9$) groups. **h** Changes in immune-related signatures between pre- ($n = 3$) and post-treatment ($n = 4$) patients. Centers, boxes, and whiskers indicate medians, quantiles, and minima/maxima, respectively, in **d-e** and **g-h**. Two-sided Wilcoxon rank-sum test was used for comparison in **d-e** and **g-h**. Source data are provided as a Source Data file.

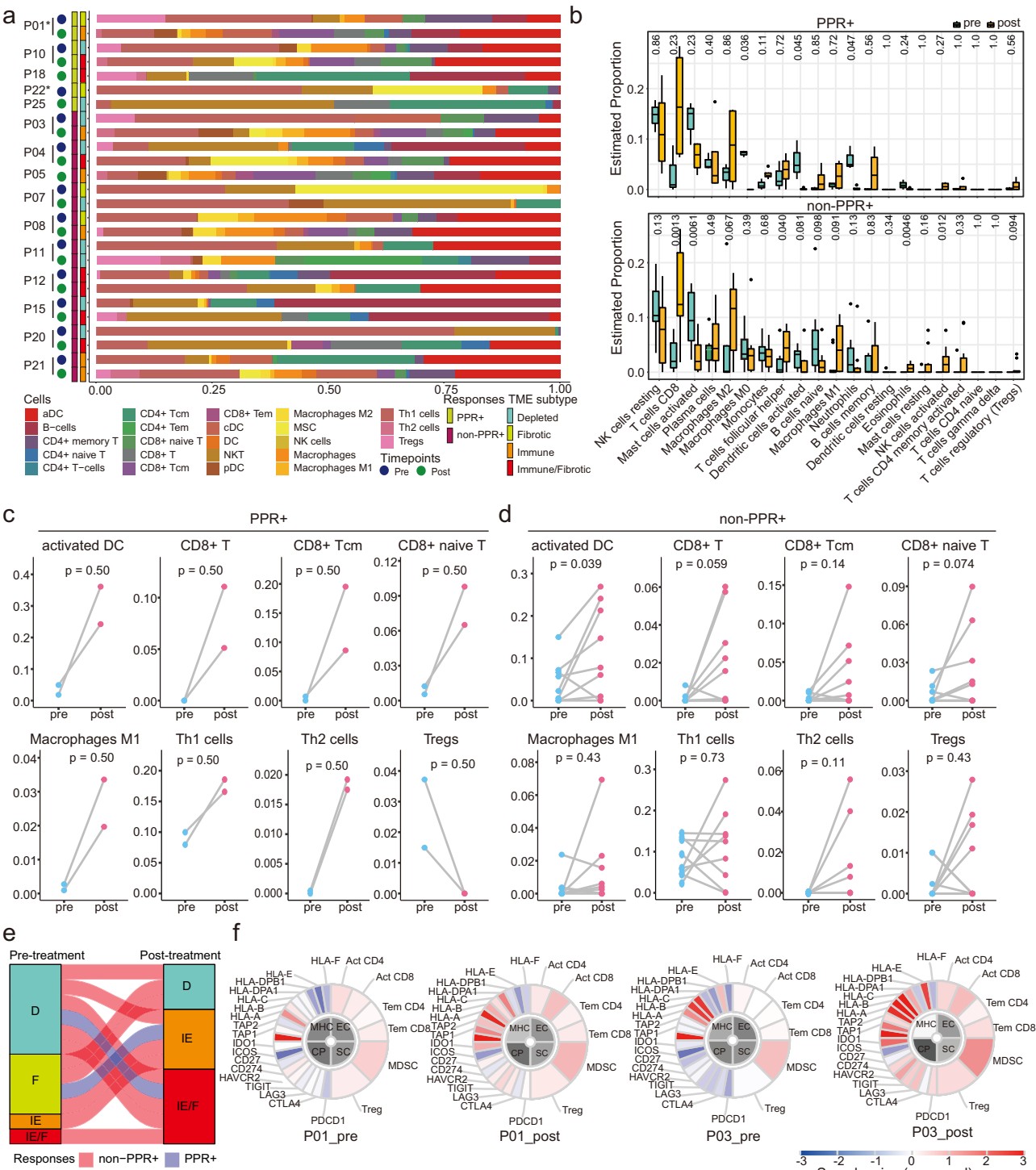

**Fig. 5 | Immune cell deconvolution and association with responses to neoadjuvant therapy. a** Abundance of infiltrated cells estimated by xCell in 15 patients. **b** Changes in proportions of 21 immune cell types from pre- to post-treatment samples in PPR+ (top, n = 3 and 4) and non-PPR+ (bottom, n = 9 and 10) patients, estimated by Cibersort. Centers, boxes, whiskers, and dots indicate medians, quantiles, minima/maxima, and outliers, respectively. Two-sided Wilcoxon rank-sum test was used with no adjustments for multiple comparisons. **c-d** Changes in proportions of immune cells from pre- to post-treatment samples in PPR+ (**c**, n = 2)

and non-PPR+ groups (**d**, n = 9). Two-sided Wilcoxon signed-rank test was used for comparison in **c-d**. **e** Changes in TME subtypes pre- to post-treatment. Tumors were classified into four subtypes, including fibrotic (F), depleted (D), immune-enriched (IE), and immune-enriched/fibrotic (IE/F) types. **f** Immunophenotypes of patients P1 and P3. The major determinants are involved in four categories: MHC molecules (MHC), immunomodulators (CP), effector cells (EC), and suppressor cells (SC). Source data are provided as a Source Data file.

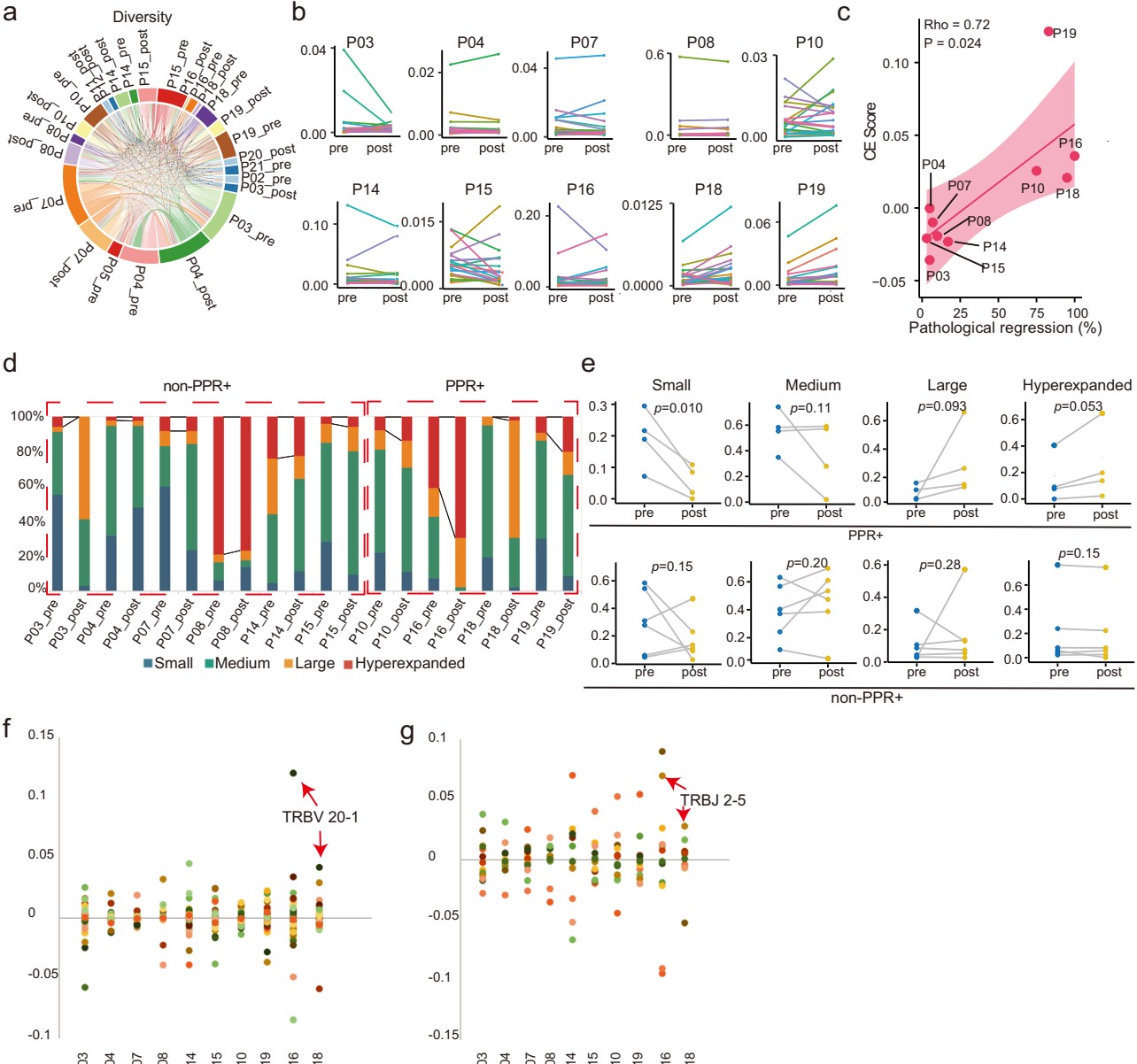

**Fig. 6 | T cell repertoire and dynamic changes. a** Pairwise overlap circos plot showing total numbers of clonotypes shared between samples in 16 patients. **b** Abundances of top 20 most frequent TCR clonotypes in each patient pre- and post-treatment. **c** Correlation between clone expansion scores (CE scores) and percentages of pathological regression. The correlation was accessed by Spearman's rank correlation coefficient (rho). The regression line is red and the shading indicates the 95% confidence interval. **d** Proportion of four TCR clonotypes classified by their frequency (hyperexpanded, >0.01 and ≤1; large, >0.001 and ≤0.1; medium, >0.0001 and ≤0.001; small, ≤0.0001). Black lines highlight changes in proportions of hyperexpanded clones. **e** Changes in four TCR clonotypes from pre- to post-treatment in PPR+ (top, $n = 4$) and non-PPR+ (bottom, $n = 6$) patients. Two-sided Wilcoxon signed-rank test was used for comparison. **f-g** Differences of TRBV (**f**) and TRBJ (**g**) segment usages from pre- to post-treatment samples. Source data are provided as a Source Data file.

Consistent with this, adding apatinib to conversion chemotherapy showed no anastomotic leakage or wound-healing complications in patients with advanced gastric cancer[42].

By comparing baseline pathology and omics data between patients with different responses, we identified potential biomarkers associated with pathological responses. The established biomarkers in advanced cancers[43,44], PD-L1 positive, MSI-H, and TMB-H, were present in 60–80% of MPR patients. In addition, *RREB1* and *SSPO* mutations showed comparable or improved association with MPR. *SSPO* mutation, *RREB1* mutation, and TMB-H were observed in PD-L1-negative MPR patients. In particular, the combination of PD-L1 expression and *RREB1* mutation was present in 100% of MPR patients and in none of

non-MPR patients (Supplementary Table 3). *SSPO* is a pseudogene in humans with an unknown role in cancer[21]. RREB1 is a RAS transcriptional effector and mediates TGF-β-activated EMT in cancer[45]. The biological mechanisms under the association between *SSPO* or *RREB1* mutations and responses are unclear and remain to be determined. Further, transcriptome shows that patients that responded well had "hot" tumors with IFN-γ expression, cytolytic signatures, and PD-L1 expression, consistent with prior reports[46]. These "hot" tumor features may be partially due to the MSI-H patients, which comprised two-thirds of responders[47]. By contrast, baseline TCR sequencing gives no clues to predict pathological response. Of note, all the biomarkers are putative, without validation in other cohorts yet.

By comparing omics data pre- and post-treatment, we revealed the dynamic evolution of tumor subclones, TME, and T cell repertoires during the neoadjuvant treatment. Similar to the description of therapy-induced clone evolution[48], mutations in tumors with responders predominated with "lost", while many "persistence" and "gain" mutations existed in patients with poor responses. Diminished subclones were observed in most tumors, but persistent or new dominant subclones might rise in patients with poor responses, suggesting that our neoadjuvant therapy shifted the landscape in favor of specific tumor subclones[49]. Transcriptome analysis showed that the neoadjuvant immune-based therapy fully activated the PPR+ TME by significantly upregulating distinct immune cell subsets, such as DC cells, CD8+ T cells, and polarized macrophages, consistent with previous reports in melanoma[49]. Interestingly, this type of immune activation occurred in tumors both with or without responses. CD8+ T cell infiltration was visualized in an MPR and two non-MPR patients by multiplex immunofluorescence. Enrichment of macrophages was also observed in two of them. By contrast, immune-suppressing cells, including MDSC and $T_{reg}$ cells, were divergent between responders and non-responders; these cells remarkably expanded in many non-PPR+ tumors by cell estimation, and enrichment of $T_{reg}$ cells was visualized in a non-MPR patient by immunofluorescence. $T_{reg}$ cells and MDSC are key players in sustaining an immunosuppressive TME[50,51] and are responsible for ICI resistance[52,53]. In a recent study, chemotherapy also demonstrated recruitment of CD8+ T cells and M1 macrophages in responders in the first-line treatment of advanced gastric cancer, but B cells, other than MDSC and $T_{reg}$ cells, were increased in non-responders[54]. However, our results were limited because the immune cells were deconvoluted from bulk transcriptome data, and sample sizes presumably underpowered the differences. Further, PD-L1 expression became positive in 4 non-responders after treatment. Although these therapy-induced "hot" tumors did not respond to neoadjuvant therapy, the resultant PD-L1 expression may benefit from postoperative adjuvant immunotherapy.

Dynamic expansion of T cell clones in peripheral blood, rather than the baseline levels, is closely associated with pathological remission. Our CE score accurately differentiated the 4 responders from the 6 non-responders. With more feasibility and convenience than a tumor biopsy, peripheral blood TCR sequencing might be of great value in predicting the efficacy of neoadjuvant therapy or even differentiating pseudoprogression from true progression. Meanwhile, *TRBV20-1* and *TRBJ2-5* were amplified in both MSS patients with MPR, suggesting that these two segments might have anti-tumor activity, and their expansion might be predictors of response to neoadjuvant immunotherapy in MSS gastric cancer. Although the TCR analyses were only from peripheral T cells, they could partially represent the tumor-infiltrating cells since neoantigen-specific T cells can be identified in peripheral blood[55].

There are several limitations in this study. Due to its exploratory nature, the sample size is small without controlled arms. Several phase III trials on ICI-based neoadjuvant therapy are recruiting, such as KEYNOTE-585 (ClinicalTrials.gov Identifier: NCT03221426) and DRAGON-IV/Ahead-G208 (ClinicalTrials.gov Identifier: NCT04208347). Moreover, the multi-omics analyses might be disturbed by sampling sites and tumor purity. Large multi-omics studies are necessary to define the best predictive biomarkers of pathological responses and OS in neoadjuvant therapy. In addition, the single-arm design of this combination therapy prevents us from distinguishing the relative contributions of each component (ICI, apatinib, and chemotherapy) on treatment efficacy and immune activation. Finally, the patients in this study received relatively short durations of treatment, including the number of treatment cycles and the interval between the last dose of apatinib and surgery, which may be inadequate for a full materialized immune response, especially for the cT4 disease. Prolonging treatment duration may improve outcomes, but the optimal strategy is to be investigated.

In conclusion, our data suggest that the ICI- and antiangiogenesis-based neoadjuvant/conversion therapy has good efficacy and feasibility in cT4a/bN+ gastric cancer, especially the MSI-H and PD-L1 positive patients. How to improve its efficacy in MSS and PD-L1 negative patients needs further exploration. The multi-omics findings provide some candidate efficacy-related biomarkers and help us understand the mechanisms of the treatment responses and resistances.

## Methods

This study is an investigator-initiated, phase II, single-armed trial in a single institution. It was approved by the Medical Ethical Committee of Shandong University Qilu Hospital (Number: 2018214) and was conducted in accordance with the Declaration of Helsinki. This clinical trial was registered at https://www.clinicaltrials.gov before patient enrollment (clinical trial identifier NCT03878472).

### Patients

Eligible patients were 18–70 years old and had clinical T4a/bN+M0 gastric adenocarcinoma, according to the 8th edition of the AJCC Cancer Staging System. Clinical stages were assessed by physical examination and contrast-enhanced CT of the neck, chest, abdomen, and pelvis. A total of 25 patients were enrolled, and all patients provided written informed consent. Patients P1 and P18 have confirmed their approval of CT and pathological images in this article. The first patient was enrolled on May 18, 2019, and the last was recruited on August 25, 2020.

### Trial design and treatments

Eligible patients received at least two cycles of camrelizumab (200 mg d1), apatinib (250 mg qd d1-14), and S-1 (50 mg bid d1-10) with or without oxaliplatin (85 mg/m² d1) every 2 weeks (Fig. 1A). Then patients were re-evaluated and underwent surgery after apatinib withdrawal for at least 14 days. Patients did not receive laparoscopes before the neoadjuvant treatment. Pre- and post-treatment tissues were collected by gastroscope and surgery for immunohistochemistry, multiplex immunofluorescence, WES, and transcriptome sequencing (Fig. 1A). Peripheral blood was collected for routine lab examination and TCR sequencing. Adverse events were evaluated according to Common Terminology Criteria for Adverse Events version 5.0. Postoperative complications were evaluated according to the Clavien-Dindo classification[56]. Radiological responses were evaluated according to Response Evaluation Criteria in Solid Tumors version 1.1. The primary endpoint is pathological responses and their potential biomarkers. Secondary endpoints included safety, objective response, 1-year PFS rate, and 1-year OS rate. Analyses of Becker TRG, mIHC, Immunophenoscore, and TME subtypes were performed post-hoc. Clinical data were organized in Microsoft Excel version 2019.

### Pathological assessments

Patients who received per-protocol treatment and tumor resection were evaluated for pathological responses. Surgical specimens were stained with hematoxylin and eosin and analyzed by pathologists for the percentage of residual viable tumor cells in tumor beds. Complete pathological response (CPR) was defined as no viable tumor cells. Major pathological response (MPR) was defined with no more than 10% viable tumor cells. Partial pathological response (PPR) and PPR+ were defined as no more than 50 and 30% viable tumor cells, respectively. In addition, tumor regression was also classified using the Becker tumor regression grading (TRG) system[32,57], which includes the following

categories: TRG1a (no residual tumor cells), TRG1b (<10% residual tumor cells); TRG2 (10–50% residual tumor cells); and TRG3 (50% or more residual tumor cells). Immunohistochemical staining or in situ hybridization was performed to evaluate PD-L1, *ERBB2*, and mismatch repair (MMR) proteins (MLH1, MSH2, MSH6, and PMS2). Fluorescence in situ hybridization was used for ERBB2 (2+) samples.

## Whole-exome sequencing and read alignment

WES was implemented on the formalin-fixed paraffin-embedded tumor tissue and matched peripheral blood samples. GeneRead DNA FFPE Tissue Kit (QIAGEN, GER) was employed for FFPE section extraction, while Mag-Bind® Blood & Tissue DNA HDQ 96 kit (OMEGA) was utilized for blood sample extraction. The dsDNA HS Assay Kit (ThermoFisher Scientific, USA) was used for DNA quantification. Sequencing libraries were built by SureSelect XT Human All Exon V6 (Agilent), and sequencing procedures were utilized by the NextSeq 550AR platform with 150-bp paired-end reads. SOAPnuke[58] was implemented to cut adapters and remove low-quality raw reads. Clean reads were aligned against the human reference genome (hg19) with BWA (v0.7.12)[59], and duplicated reads were removed by Sambamba (v0.5.4)[60]. Subsequently, generated BAM files were used for downstream analysis.

## Somatic variant calling

We compared tumor and matched blood sequencing data to identify the somatic mutations, including single nucleotide variants (SNVs) and small insertions and deletions (Indels), by 3 different mutation callers (Varscan v2.4[61], MuTect2[62], and Strelka v2.9.10[63]) with default parameters. Three callers were run with dbSNP (version 147)[64], 1000 G (phase3_release_v5)[65], CLINVAR (version 151) and COSMIC (version 81)[66] data for known polymorphic sites. Substitutions and indels with low variant allelic fractions (VAF < 0.02) or low read coverages were filtered out. Mutations called by at least 2 callers were retained. In addition, the filtered mutations were annotated by snpEff (v4.3) with NCBIrefseq (https://www.ncbi.nlm.nih.gov/refseq/).

## Tumor mutational burden, tumor neoantigen burden, microsatellite instability, and HLA

TMB was defined as the number of nonsynonymous somatic mutations per megabase. TNB was determined as previously described[67]. Briefly, HLA typing of tumor and paired blood samples were determined by POLYSOLVER (v1.0)[68] and Bwakit (v0.7.11)[59] from WES data. Secondly, somatic mutations were translated into 21-mer peptide sequences by an in-house script centered on the mutated amino acid. A sliding window approach was applied to create a 9-11-mer peptide for MHC class I binding affinity prediction. Thirdly, NetMHCpan[69] was performed to calculate the MHC affinity based on HLA type and selected peptides. Fourthly, the peptides with IC50 < 500, representing a strong binding affinity to the patient-specific HLA allele, were considered neoantigens. Eventually, TNB was evaluated as the number of neoantigens examined per megabase. Microsatellite instability (MSI) was called by MSIsensor (v0.6)[70]. The TCGA Stomach Cancer (STAD) and TCGA Pan-Cancer (PANCAN) data were used for survival analysis on UCSC Xena Browser (https://xenabrowser.net).

Tumor purity was estimated computationally from WES data of all samples using allele-frequency-based imputation of tumor (All-FIT)[71]. To determine whether the TMB loss was only due to purity drop, we performed a simulation to balance the purity between pre- and post-treatment samples. In 7 pairs of samples with large purity differences (> 1.5 times), the WES data of relatively high-purity samples were extracted randomly and then blended with the matched control samples to get the same purity as their paired tumor samples. The resultant targeted sequencing depth was considered to ensure the targeted sequencing depth of mixed samples was the same as the original sample. Then, somatic mutations were re-called according

to the same procedure as above, and TMBs were re-calculated accordingly.

## Tumor clonality and clonal genes

PyClone (v0.13.1)[72] was used to estimate the number of clones and calculate the cellular prevalence of inferred mutational clusters. For all patients with both pre- and post-treatment samples, if the change of mutation VAF was consistent with its clone's cellular prevalence, the host gene of the mutation was identified as a clonal gene.

## Whole transcriptome sequencing

Total RNA of tumor samples was isolated using RNeasy Plus Universal Kits (Qiagen, GER). RNA concentration was quantified using Qubit™ RNA HS Assay Kit (ThermoFisher Scientific, USA). RNA purity and integrity were analyzed using Take3 (BioTek, USA) and the RNA Cartridge kit of the Qseq100 Bio-Fragment Analyzer (Bioptic, CHN), respectively. Then, RNA-seq libraries were constructed using VAHTS mRNA-seq V3 Library Prep Kit for Illumina (Vazyme, CHN). Libraries were sequenced on the NextSeq 550AR platform with 150 bp paired-end reads. Quality control of WES data was described in Supplementary Data 3.

## RNA-Seq raw data quality control and gene expression analysis

Raw RNA sequencing data from the sequencer were processed to filter out low-quality reads. Clean reads from each sample were obtained and used for the following analysis. Read counts and transcripts per million values were calculated based on pseudoalignment of RNA sequencing reads to reference transcripts downloaded from GENCODE (v38) database, as implemented in Kallisto (v0.46.2)[73]. Then, gene expression levels were summarized from transcript levels. Differential expression genes (DEG) were identified by the DESeq2 package[74]. The genes with fold changes >4 or <1/4 and *P*-value <0.01 were considered as DEGs. Volcano plots and heatmaps were drawn in R with ggpubr and Complexheatmap package. Gene Ontology (GO) and Kyoto Encyclopedia of Genes and Genomes (KEGG) pathway enrichment were analyzed by the KOBAS-i webtool KOBAS[75].

## Infiltration abundance of immune cell

Based on the gene expression matrix, R packages including xCell[76] and Cibersort[77] were used to estimate the infiltration abundance of immune cells for each sample. T-Cell-Inflamed Gene-Expression Profile (GEP)[78] was employed to evaluate the tumor immune signatures comprised of cytolytic, IFN-gamma, T-cell, Batf3-DC, and HLA. Immunophenogram (https://tcia.at/home)[79] was implemented to assess immunophenotypes which consist of MHC molecules (MHC), Immunomodulators (CP), Effector cells (EC), and Suppressor cells (SC).

## TME subtyping

TME subtyping was performed according to a previous publication[23], which defined four TME subtypes: "depleted", "fibrotic", "immune-enriched/non-fibrotic", and "immune-enriched/fibrotic", for pan-cancer RNA-seq data. RNA-seq data from our study were classified by a KNN model trained by TCGA STAD samples using an R package CLASS.

## TCR sequencing

For TCR sequencing, total RNA was isolated from peripheral blood mononuclear cells (PBMCs) by RNeasy Plus Mini Kit (Qiagen, USA) according to the manufacturer's instructions. Take3 (BioTek, USA) was applied to determine the final concentration. Total RNA was synthesized into the cDNA library by iRepertoire Short Read iR-Profile Reagent System HTBI-vc. Sequencing was performed by NextSeq 550AR platform with 150-bp paired-end reads. Fastq reads were trimmed based on their low-quality 3′ ends bases. Trimmed pair-end reads were integrated according to overlapping alignment with the modified

Needleman-Wunsch algorithm. MiXCR (v2.1.10)[80] was used to identify the CDR3 sequences of V-D-J gene segments with reference sequences from the IMGT[81]. VDJtools (v1.2.1)[82] was utilized to assess the immune repertoire sequencing. A frequency-based correction was performed on samples. The Shannon and D50 indexes were used to estimate the diversity of the TCR clone. Based on the top 20 most frequent TCR clonotypes, a clone expansion score (CE score) was defined as the sum of the differences in clonotype abundance between pre- and post-treatment. According to clonotype abundance, TCR clonotypes were classified into four groups, including hyperexpanded (frequency >0.01 and ≤1), large (>0.001 and ≤0.1), medium (>0.0001 and ≤0.001) and small (≤0.0001).

### PD-L1 IHC
PD-L1 IHC was performed using the PD-L1 IHC 22C3 pharmDx kit (Dako) on the Dako ASL48 platform according to manufacturer recommendations. HER2 IHC was performed using the HER2/neu kit (Ventana) following the standard preprogrammed staining protocol. The anti-PD-L1 antibody (clone: 22C3) and anti-HER2 antibody (clone: 4B5) were provided already diluted at an unspecified ratio in the kit.

### Multiplex immunofluorescence
Selected samples were assessed by multiplex immunofluorescence with antibodies against cytokeratin (Zsbio, clone number: AE1/AE3, 1:200 dilution), CD8 (Abcam, clone: C8/468 + C8/144B, dilution: 1:200), FoxP3 (Abcam, clone: 236A/E7, dilution: 1:100), CD68 (Abcam, clone: KP1, dilution: 1:100), PD-1 (Zsbio, clone: UMAB199, dilution: prediluted), and PD-L1 (Cell Signaling Technology, clone: E1L3N, dilution: 1:200). The staining was performed using the Opal 7-Color IHC Kit (Akoya Biosciences, USA) and imaged by a PerkinElmer Vectra 3.0 (Perkin Elmer, Hopkington, MA) multispectral microscope. Specificity for each staining has been validated.

### Statistical analysis
All statistical analyses were implemented by R 3.6.1 software. There is no prespecific endpoint or criteria for sample size. Medians and quartiles were provided for distributions of time intervals. 95% CIs were constructed using the Clopper-Pearson method for pathological response rates. The Wilcoxon rank-sum test was used to compare TMB, TNB, immune-related gene expression, signature levels, and immune cell estimations between independent groups. The Wilcoxon signed-rank test was used to compare TMB, TNB, immune-related gene expression, signature levels, immune cell estimations, and T cell clone frequencies between matched samples (pre- vs. post-treatment). The Fisher's exact test evaluated associations of pathological responses with gene mutation, TMB, TNB, MSI, and PD-L1 status. Correlations between pathological regressions and CE scores or lost SNV were assessed by the Spearman's rank correlation coefficient. OS and DFS were estimated using the Kaplan-Meier method, and differences between groups were assessed by the log-rank test. All reported $P$ values are two-sided, and $P$ values less than 0.05 were considered statistically significant.

### Reporting summary
Further information on research design is available in the Nature Portfolio Reporting Summary linked to this article.

## Data availability
WES, transcriptome sequencing, and TCR sequencing data generated in this study have been deposited in the Genome Sequence Archive under the accession code HRA002181. The sequencing data are available under controlled access due to data privacy laws related to patient consent for data sharing and the data should be used for research purposes only. Access can be obtained by approval via the Data Access Committee in the GSA-human database (for further instructions, please refer to: https://ngdc.cncb.ac.cn/gsa-human/document/GSA-Human_Request_Guide_for_Users_us.pdf). The approximate response time for accession requests is about 4 weeks, and access is granted for one year.

Mutation data of the TCGA Stomach Cancer and TCGA Pan-Cancer cohorts were from the UCSC Xena [https://xena.ucsc.edu/]. The reference datasets included GENCODE (v38) database [https://www.gencodegenes.org/human/], dbSNP (version 147) [https://www.ncbi.nlm.nih.gov/snp], 1000G (phase3_release_v5) [https://www.internationalgenome.org], CLINVAR (version 151) [https://www.ncbi.nlm.nih.gov/clinvar], COSMIC (version 81) [https://cancer.sanger.ac.uk], and IMGT [https://www.imgt.org/].

CT scan and pathological imaging are not shared due to patients' privacy. The other individual de-identified participant data, Study Protocol, and Statistical Analysis Plan are available on reasonable request within 3 years after this paper's publication. Qualified researchers may request access to individual patient-level clinical data by contacting the corresponding author at lianliu@sdu.edu.cn. The remaining data are available within the Article, Supplementary Information, or Source Data file. Source data are provided with this paper.

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

## Acknowledgements

We thank all patients and their relatives for being a part of the trial. Hengrui Pharmaceuticals supplied Camrelizumab. We thank Drs. Ming Lu, Shaowei Sang, Xiaorong Yang, Yuan Zhang, Hao Chen, and Tongchao Zhang in the Clinical Research Center of Shandong University for their assistance in statistical analysis. We thank Alexandra H. Marshall (Marshall Medical Communications) for editing this manuscript. This work was supported by the National Natural Science Foundation of China (82173305, L.L.), Natural Science Foundation of Shandong Province (ZR2017MH005, L.L.), and Foundation of Shandong University Clinical Research Center (2020SDUCRCC011, S.L.). The funders had no role in study design, data collection and analysis, or manuscript writing.

## Author contributions

L.L. conceived the study. F.X. wrote the protocol with input from S.L. and L.L. L.L., W.Y., and S.L. were responsible for patient treatment and patient care with help from Q.X., X.Z., M.H., K.H., X.D., Jian W., and J.L. L.L., W.Y., Z.L., C.C., and M.W. were responsible for patient recruitment. S.L., W.Y., and F.X. were responsible for coordinating trial procedures and collecting data and samples. S.L., W.Y., and L.L. were clinical investigators. W.Y., Z.L., M.W., and C.C. were responsible for surgery. S.L. analyzed and interpreted clinical data with support from L.L. and L.S. W.L. and D.Y. helped with clinical imaging. D.S. and P.G. helped with pathological evaluation. S.L. and L.S. performed bioinformatic analysis and statistical analyses with help of H.L., W.Z., Jiaqian W., and Z.Z. S.L. wrote the manuscript with input from L.L. All authors reviewed the manuscript, interpreted data, and approved the final version.

## Competing interests

H.L., W.Z., J.W., and Z.Z. are employees at Yucebio Technology. The remaining authors declare that they have no conflict of interest.

## Additional information

[1]Department of Medical Oncology, Qilu Hospital, Cheeloo College of Medicine, Shandong University, Jinan 250012 Shandong, China. [2]Department of General Surgery, Qilu Hospital, Cheeloo College of Medicine, Shandong University, Jinan 250012 Shandong, China. [3]Department of Pharmacy, Qilu Hospital, Cheeloo College of Medicine, Shandong University, Jinan 250012 Shandong, China. [4]Shenzhen Yucebio Technology Co., Ltd., Shenzhen, 518000 Guangdong, China. [5]Department of General Surgery, Zibo Municipal Central Hospital, Binzhou Medical College, Zibo 255036 Shandong, China. [6]Department of Radiology, Qilu Hospital, Cheeloo College of Medicine, Shandong University, Jinan 250012 Shandong, China. [7]Department of Pathology, Qilu Hospital, Cheeloo College of Medicine, Shandong University, Jinan 250012 Shandong, China. [8]Department of Medical Oncology, Shandong Provincial Hospital of Traditional Chinese Medicine, Jinan 250012, China. [9]These authors contributed equally: Song Li, Wenbin Yu. ✉e-mail: lianliu@sdu.edu.cn

