## [Peer Review File · Nature Communications]

Neoadjuvant therapy with immune checkpoint blockade, antiangiogenesis, and chemotherapy for locally advanced gastric cancerREVIEWER COMMENTS

Reviewer #1 (Remarks to the Author): with expertise in gastric cancer, clinical

Summary: In their submitted report, Li and colleagues describe an investigator-initiated, phase II single-arm trial performed at a single institution studying the efficacy and safety of camrelizumab, apatinib and S-1 +/- oxaliplatin. In their study, they also include rich correlative work utilizing several complementary techniques such as exome/transcriptome sequencing and TCR analysis. Their primary findings include demonstrable safety and efficacy of the aforementioned neoadjuvant regimen and detailed analysis of how this treatment influences TMB, TNB, T cell clonal expansion and cellular composition/transcriptional signatures of the tumor microenvironment. The manuscript is overall well organized and clearly written. Limitations include some aspects of the trial design (small sample size, lack of control arm) and resulting limitations in power for correlative studies and inability to dissect relative contributions from each treatment component (chemotherapy versus checkpoint blockade versus anti-angiogenic therapy). I would ask the authors to address the following comments:

Abstract/Introduction:

1. Overall, clearly written. Some spelling and grammatical errors are noted in the minor points section below.

Methods:

1. Was HER2+ status an exclusion criteria in the trial?
2. How is "molecular biomarkers" a primary end point of the trial? Please provide a statistical section to support the power calculations and sample size determination for the trial.
3. A standardized pathologic assessment system such as Becker or Mandard should be used for path assessment. What system was used here? This is important to increase comparability to other trials which the authors do in the discussion.

Results:

1. Table 1: It seems that PD-L1 CPS is shown but would be nice for this to be explicitly stated.
2. Line 83-84: additional detail on how many doses and what neoadjuvant treatments exactly were administered would be beneficial. It would be helpful to depict these details graphically or perhaps as a supplementary table. These details are especially important given the small sample size and therefore increased importance of each patient's therapeutic context.
3. With such a small sample size, inclusion of 4 dMMR patients risks a dominant driver effect in efficacy from this minority fraction of patients. It would be helpful to have analyses presented with dMMR/MSI-H patients excluded so as to better understand the magnitude of benefit in pMMR/MSI-low patients. For example, in Supp Fig 3, are there still significant pre/post-treatment changes in TNB and TMB if P01 is excluded?
4. Since pathologic response is a primary outcome, movement of Table S1 into the main manuscript should be considered, especially pathological complete response rate as this outcome helps interpretation/comparison to other landmark trials such as FLOT4-AIO.
5. Figure 2A: Tumor fraction/purity should be reported along with TMB so as to increase clarity. As presented, the data risks suggesting that TMB in the same tumor cells decreases after treatment which is incorrect.
6. The findings in Figure 2F for pre/post microenvironment changes in cellular composition is interesting and well demonstrated. A side comparison in a patient with non-MPR would strengthen this argument.
7. I would recommend softening the statement regarding co-localization of macrophages and tumor cells equating to phagocytosis as an anti-tumor mechanism, further mechanistic work is needed to definitively state this.
8. Regarding results shown in 2G and 2H. Why do the authors think we see these trends? Do they believe this is from immunoediting?
9. Supp Fig S2-can the authors define RCCEP? Could include in legend.
10. Could the authors comment on how these data inform optimal time to surgery? This is important as we begin to think about the temporal dynamics of an optimal anti-tumor immune response in the

neoadjuvant setting. It is possible that with a mean of 16 days and a tight range (14-19 with one exception), there was not adequate time to capture a full materialized immune response. Similarly, there is a possibility that a longer duration of neoadjuvant therapy would improve outcomes. In fact in clinical T4 patients one could argue for a more extended duration of chemotherapy such as in the FLOT4 trial. This warrants comment

11. Supp Fig 3: I don't understand 3c that shows no significant difference in pre/post change of TMB, TNB and MSI yet the statement in the manuscript is that there is a significant change (line148-149)

12. Line 179-181: Reads a bit awkward as currently phrased. Suggest breaking into two sentences for increased clarity.

13. Line 206-209: The statement that neoadjuvant therapy promoted cytolytic and CD8+ effector signatures in all patients (4H and Supp fig 8) should be modified to clearly state that these differences were not statistically significant (presumably due to being underpowered)

14. Figure 5: Limited power should be clearly stated in limitations

15. Figure 5D: If available, it would be nice to see this responder/non-responder anecdotal comparison but for the aggregate cohort instead of selected single patients. This could be included in the supplement.

16. Figure 5A could be depicted more clearly. For example, left of patient ID #, PPR+ vs non-PPR+ status could be shown.

17. Figure 6: Is anything known in the literature about the specific VDJ sequences observed?

18. Figure 6: Increased clarity would be appreciated regarding whether the entire analysis is from peripheral T cells or if TCR analysis using tumor tissue was used as well.

Discussion:

1. A limitation of this small (but data dense) study is that characterization of the relative contribution of each therapeutic component ie: chemotherapy versus checkpoint inhibitor versus anti-angiogenic agent is lacking

2. The multiplex immunofluorescence work was quite limited and descriptive in nature. If additional data is available, it would have beneficial to include with quantification of changes in cellular composition such that comparison could be made with the results of cellular deconvolution from bulk transcriptome data. Further, beyond the suggestion that macrophages co-localize with tumor cells, there is no discussion of tumor and tumor microenvironment dynamics in the spatial dimension and additional multiplex IF data could potentially explore this. An representative study (albeit focused on colorectal cancer) is Bortolomeazzi et al, *Gastroenterology* 2021; 161:1179-1193

3. Line 260-261: This statement should be softened because pathologic complete response is comparable to phase III data from FLOT4.

4. A limitation of this study is relatively short follow up (~15 months). It remains unclear if benefit in more longitudinal outcomes DFS, PFS, OS is seen which is highly relevant in the context of increase anti-tumor immunity which would theoretically decrease distant metastatic recurrences.

5. Line 270-272: This is an overstatement given that multiplex IF was only shown for 1 patient.

6. The authors should be commended for including a high risk population (cT4N+) in their study.

7. The discussion should include commentary on how the putative biomarkers explored in this study have the potential to outperform PD-L1 for predicting response to checkpoint blockade. Have the authors offered data to support a superior marker? For example, does TCR and WES data add any predictive power beyond PD-L1 status in identifying pathologic responders. This is an important question for the field and it is possible that PD-L1 expression is the end result of the favorable features (high TMB, etc) that the authors identify here.

8. The overall discussion needs to be tempered to avoid overstating the impact of the data. There is for example no evidence that bringing immunotherapy into the neoadjuvant setting in gastric cancer is superior to chemotherapy

9.

Minor Points:

Abstract:

1. Line 30: suggest swapping unsatisfactory to “limited” or “poor”
2. Line 30: There is a typo in “antiangiogenic”
3. Line 30: suggest swapping “effect” for activity
4. Line 33-34: The treatment arms were not introduced clearly yet and thus it is unclear to which treatment arm the path response rates pertain to.
5. Line 35: suggest removing ‘s’ in “tumor mutational burdens”
6. Line 39: suggest “...during neoadjuvant immunotherapy.”
7. Line 41: suggest deleting “efficacy-related”

Introduction:

1. Line 62-63: Suggest swapping “was an attractive and critical question” for “...remains incompletely explored.”
2. Line 64: typo in “anti-angiogenetic” (the manuscript should be checked for this typo as it occurs in several additional places beyond line 64).
3. Line 64: Inclusion of a hyphen ie: Anti-angiogenic versus antiangiogenic should be decided upon and the manuscript should be uniform
4. Line 75: typo in “whole-exon”, should be “whole exome sequencing”

Reviewer #2 (Remarks to the Author): with expertise in cancer genomics

Li et al. have evaluated the efficacy of ICI and antiangiogenic agents in late-stage gastric cancer in the neoadjuvant setting, and further investigated the genomic, transcriptomic and immune-related features associated with pathogenic response. Overall, the topic and approach of the manuscript is interesting and should be of interest to Nature Communications readers. However, I have several major concerns, especially on the WES analysis and the reported results on SSPO mutations.

1. I could not reproduce the reported association between SSPO (SSPOP) mutations and overall survival in the TCGA STAD cohort. SSPO is mutated in only 3/478 (0.6%) patients in the STAD cohort based on data from cBioPortal, and there is no association ($P=0.3$) between SSPO mutations and overall survival:

https://www.cbioportal.org/results/oncoprint?cancer_study_list=stad_tcga&Z_SCORE_THRESHOLD=2.0&RPPA_SCORE_THRESHOLD=2.0&profileFilter=mutations&case_set_id=stad_tcga_sequenced&gene_list=SSPOP&geneset_list=&tab_index=tab_visualize&Action=Submit&comparison_subtab=survival

1. I noticed that the mutation calling was performed with VarScan using a VAF cutoff of 0.02 (2%). The authors did not specify the sequencing coverage of the WES samples. Assuming 100x coverage, a position with VAF=0.02 translates to only 2 supporting reads. At such a low VAF cutoff, the mutations called would have very low precision with a large number of false positives. Furthermore, recent benchmarking papers found VarScan to perform poorly compared to other mutation callers (Wang et al. SomaticCombiner: improving the performance of somatic variant calling based on evaluation tests and a consensus approach. *Sci Rep* 10, 12898 (2020). <https://doi.org/10.1038/s41598-020-69772-8>; Huang et al., SMuRF: portable and accurate ensemble prediction of somatic mutations, *Bioinformatics*, Volume 35, Issue 17, 1 September 2019, Pages 3157–3159, <https://doi.org/10.1093/bioinformatics/btz018>). I would recommend that the authors use an additional best-in-class mutation caller (such as Mutect2 or Strelka2) to produce high confidence mutation calls (e.g. taking the intersection of Mutect2/Strelka2/VarScan).
2. The authors identified the SSPO gene to be frequently mutated in MPR patients, but not in non-MPR patients. What is the functional role of the gene/protein (it is annotated as a transcribed pseudogene by Ensembl)? Where are the mutations found in the SSPO protein? Also, the figure legend for frameshift mutations seems to be wrong in Figure 3A.
3. The authors found a general decrease in TMB post treatment, and concluded that neoadjuvant therapy eliminated sensitive clones. Similarly, the authors report pathological regressions were

correlated with 'loss' of mutations and subclones (Fig. 3E/F). However, the authors have omitted to explore whether the decrease in the number of mutations detected could be explained by reduced tumor purity (viable cancer cells) post treatment. Tumor purity can be estimated with NGS/WES data. The authors should rule out that the observed 'loss' of mutations / clones / decreased TMB is not just due to a drop in tumor purity in tumors with pathological response.

4. The authors defined clonal genes as "For all patients with both pre- and post-treatment samples, if the change of mutation VAF was consistent with that of its clone's cellular prevalence, the host gene of the mutation was identified as a clonal gene". This definition can be confusing as people generally regard clonal mutations as mutations that are present in all subclones of the tumor.

5. The authors found that CD274 (PD-L1) expression decreased in post treatment samples (Figure 4E). On the other hand, they also found the number of PD-L1+ patients increased post treatment (Figure 2C). Can the authors elaborate/discuss on these seeming conflicting results?

6. P. 9 l. 181: "These genes may be candidate biomarkers for predicting responses to neoadjuvant therapy." The authors identify 293 clonal genes that either 'expand' or 'contract' in VAF upon treatment. However, it is not clear how these genes are useful as candidate biomarkers for prediction of response at baseline (before treatment).

Minor points:

1. Previous studies suggested that EBV positive gastric cancer tend to respond favorably to ICI. EBV typically comprise of around 10% of gastric cancer cases. It would be interesting to know if EBV patients also respond well in the neoadjuvant setting. Are any of the patients in this study EBV positive? If so, could the authors annotate EBV patients in Figure 2?

2. P-values should be added to Figure 1D

3. Figure legend is missing for Figure 2C

4. For Figure 4C and 4F, what do the colors of the heatmaps represent? Z-score? The figure legend should be made more descriptive.

5. The boxplots in Figure 5B show large fold change but all the p-values are not significant. Can the authors confirm the p-values are correct and perhaps show the data points on the plots?

6. Why fit a regression line through the two sets of points in Figure 3G? What's the purpose of figure 3G?

Reviewer #3 (Remarks to the Author): with expertise in gastric cancer, clinical

This manuscript describes the results of a phase II trial in which patients with resectable gastroesophageal cancer were treated with chemotherapy, a PD-1 inhibitor and an antiangiogenic TKI. All patients had locally advanced disease at presentation. Reasonably good rates of good pathological response were observed. The work's strength are the translational correlates, many of which have not been previously described in this setting.

Introduction: This is well written and highlights the specifically poor outcomes of the patients included in the current trial.

Results: Suggest changing "median observation" to "median follow up"?

CONSORT diagram – does this need to be separated into T4a vs T4b? It's a bit confusing. Suggest for the description in the paper to deal with all patients and have CONSORT reflecting this with supplementary CONSORT describing T4a and T4b?

Did patients with peritoneal mets have laparoscope pre treatment?

Of interest, 4/5 of the MPR patients were already predisposed to have a response to anti-PD-1 – only one was MSS and PD-L1 negative. This is the patient of interest. For the others, I am not sure that we can comment that antiangiogenic therapy added any value.

Post-op complications need to be reported in much more detail, especially in view of the close relationship between antiangiogenic therapy and surgery. Please report according to Clavien Dindo

and comment on ST03 and RAMSES trial results.

Is it possible to report the TMB and TNB results separately for MSS vs MSI patients? It's described already that TMB in MSI patients will decrease with anti-PD-1 (Kwon et al, Cancer Discovery 2021). It's not been done for MSS, and I suspect the result overall is driven by the MSI results.

I am not sure that the sample size is powered adequately for "gain" and "loss" analysis as described so would not place too much emphasis on this.

It's not surprising that immune inflamed genes were upregulated in responders as most of these were MSI and/or Pd-L1 positive.

The changes in the TME in responding patients have also been demonstrated in patients responding to chemotherapy (Lee et al, Cancer Discovery 2021). Can the authors comment on any added value that might have been gained from apatinib?

Discussion:

The statement that neoadjuvant ICI is superior to chemotherapy in pCR is overstated. Path CR was 25% and this is not much more than is reported with FLOT. Also the sample size is very small.

The statement that 293 genes were detected that might predict response is not valuable. Even by multiplicity some genes would be likely to be positive. This number of genes is not likely to be useful.

There is not a lot of discussion around what the apatinib added, if anything. This should be mentioned in more detail.

Overall, the conclusions need to be tempered by the fact that a large proportion of responding patients in the trial had MSI tumours and PD-L1 negative tumours (certainly MSI tumours were more than the 6-8% expected...).

Reviewer #4 (Remarks to the Author): with expertise in biostatistics, clinical trial study design

The statistical analyses have many awkward and incorrect descriptions, and analyses themselves have many flaws. The following is the list of the problems, all of which must be addressed for this report to be considered further. The statistical analysis plan must be completely rewritten with enough details so that the readers know what methods were used. It is essential that a qualified statistician handle the statistical analysis.

Please include a sample size justification.

Line 402: "Kaplan-Meier" is a name of an estimator. There is no "Kaplan-Meier analysis".

Line 470: Comparison is not estimated.

Line 471: Wilcoxon signed-rank test / pair-wise Wilcoxon signed-rank test are incorrect tests. Wilcoxon rank sum test should be used for comparison of independent groups.

Line 472: "... regression were tested" is unclear.

Line 473: "Pearson" is a name of an estimator (estimate), and it is not a name of a test.

Line 473: There is no "Log-rank regression".

Line 426: Plots are not conducted.

Throughout the manuscript, an estimate of the median is followed by a 95% confidence interval. Please describe/explain how this confidence interval is calculated. It is more common to use lower and upper quartiles to indicate data variability.

Line 425: "Log2FoldChange > 1.5" is strange. Does this mean FoldChange > 2^{1.5}? If the authors really mean "Log2FoldChange > 1.5", this cut off corresponds to 2.82 fold changes and seems

unjustified.

Figures:

Boxplots are used to show the distributions, but given small sample sizes, boxplots are not optimal ways to show data. As the authors did in some places, dots and lines connected pre- and post-observations are effective ways to show these data. (e.g. Figure S8a). Remove boxes and just show dots and lines. Additionally, dots showing differences (post - per) may be informative.

Figure S4 and its associated regression analysis are highly problematic. Because of a few outliers and influential points and abundance of zeroes, most of the fitted lines do not represent the data well. Some fitted lines go below zero. The whole analyses associated with this figure must be redone with appropriate considerations given to outliers, influential data, and zeroes. Removing them would be inappropriate.

Line 419 and other places: The word, "data", is plural.

Reviewer #1 (Remarks to the Author): with expertise in gastric cancer, clinical

Summary: In their submitted report, Li and colleagues describe an investigator-initiated, phase II single-arm trial performed at a single institution studying the efficacy and safety of camrelizumab, apatinib and S-1 +/- oxaliplatin. In their study, they also include rich correlative work utilizing several complementary techniques such as exome/transcriptome sequencing and TCR analysis. Their primary findings include demonstrable safety and efficacy of the aforementioned neoadjuvant regimen and detailed analysis of how this treatment influences TMB, TNB, T cell clonal expansion and cellular composition/transcriptional signatures of the tumor microenvironment. The manuscript is overall well organized and clearly written. Limitations include some aspects of the trial design (small sample size, lack of control arm) and resulting limitations in power for correlative studies and inability to dissect relative contributions from each treatment component (chemotherapy versus checkpoint blockade versus anti-angiogenic therapy). I would ask the authors to address the following comments:

Response: Dear Professor, we appreciate your valuable comments, which have greatly helped our study. We agree with you regarding the limitations, the low power and inability to dissect relative contributions of each treatment component, given this is an exploratory study.

When we started this trial in April 2019, there was little evidence of the efficacy of immune checkpoint inhibitor (ICI) in gastric cancer. In the late-line setting, ICI only showed efficacy in some studies, such as ATTRACTION-02 (Kang, *Lancet*, 2017) and KEYNOTE-059 (Fuchs, *JAMA Oncol*, 2018), but failed in the other studies (e.g., JAVELIN 300; Bang, *Ann Oncol*, 2018). In the second-line setting, ICI also failed (e.g., KEYNOTE-061; Shitara, *Lancet*, 2018). Plus, the trial outcomes in the first-line setting were unknown (Checkmate-649 or KEYNOTE-062). Due to the uncertain efficacy of ICI monotherapy in gastric cancer in 2019—especially in the neoadjuvant setting, risking surgery delay and inoperativeness—we designed a combined treatment regimen. We were aiming to achieve a more reliable efficacy and also to meet the ethical requirements to protect patients. To achieve a better outcome, we also adopted a VEGFR2 TKI apatinib, because it was found to synergize with camrelizumab in preclinical studies (Zhao, *Cancer Immunol Res*, 2019) and has a 16% objective response rate in refractory advanced gastric cancer (Xu, *Clin Cancer Res*, 2019). As you mentioned, the single-arm design with combined treatment could not distinguish the relative contributions of each component.

Most importantly, there were no reported trials of neoadjuvant ICI-based treatment in gastric cancer, so we were unable to estimate a pathological response rate, which is commonly used to evaluate the efficacy

of neoadjuvant treatment for sample size calculation in phase II trials. Therefore, we set a sample size for an exploratory study following similar trials in lung cancer (n = 21, Forde, *NEJM*, 2018) and melanoma (n = 20 in each arm, Amaria, *Nature Medicine*, 2018), colon cancer (n = 21 in dMMR cancer and n = 19 in pMMR cancer; Chalabi, *Nature Medicine*, 2020). To our knowledge, recent similar studies also recruited about 20 - 30 subjects (2021 ASCO abstracts, DOI: *J Clin Oncol*.2021.39.15_suppl.4040, 4026, 4046). We expect to address these issues in large controlled trials in the future.

We have made substantial amendments to the manuscript, mainly in four areas:

1. **Clinical:** We organized several meetings to discuss your comments with physicians in medical oncology, gastrointestinal surgery, pathology, and radiology. We re-explained the results that you doubted, updated the follow-up of all patients to reach a median of 24 months, and collected and reported the post-operation complications.
2. **Statistics:** We discussed the statistical questions with the statistical methodology team at our institution several times. This team is led by Dr. Ming Lu. The team helped re-perform all of the statistical analyses and proof the text.
3. **Bioinformatics:** The bioinformatics work was revised substantially. Our team re-analyzed all of the data for microsatellite stable (MSS) patients separately, got that most of these data supported previous conclusion. We used three different mutation callers to recall the mutations for more reliable results, used a simulation method to balance tumor purities for tumor mutational burden comparison, and performed Immunophenoscore analysis for all paired specimens.
4. **Multiplex immunofluorescence:** We performed multiplex immunofluorescence for additional patients with non-MPR as a side comparison.

Our changes are described in more detail below. For your convenience to review, we uploaded two manuscripts. One is with track changes, its line numbers being referred to by the response letter. A clean version was in the supplementary files, with the revisions colored in red.

Abstract/Introduction:

1. Overall, clearly written. Some spelling and grammatical errors are noted in the minor points section below.

Response: Thank you for indicating these errors. We have corrected them and re-checked spelling and grammar throughout the manuscript.

Methods:

1. Was HER2+ status an exclusion criteria in the trial?

Response: Thank you for this question. HER-2 positivity was not in the exclusion criteria. The patients with HER2+ status are shown in Table 1 or Fig. 2A.

2. How is "molecular biomarkers" a primary endpoint of the trial?

Response: Thank you for pointing this out. When designing this trial, we expected not only to evaluate the efficacy of the combined treatment, but also to focus on the underlying mechanisms and potential biomarkers, providing preliminary clues for further stratification analysis or mechanistic studies. However, without thorough consideration, we wrongly assigned "molecular biomarkers" as the primary endpoint. We apologize for this inappropriate description, which is misleading and confusing. As you said, we should correct this mistake in the manuscript. In fact, the "biomarker" was reported in the way of exploratory data in the manuscript, mainly referring to the association between biomarkers and pathological responses. Because biomarkers were not used for sample size calculation, it does not influence the conclusion. Through repeated discussions with statisticians at our institute, we decided to set the "biomarkers" as the exploratory endpoint and made relative revisions (**Methods, lines 436 - 437**).

3. Please provide a statistical section to support the power calculations and sample size determination for the trial.

Response: We appreciate this comment. We agree that a sample size calculated based on an assumption of efficacy is better for this study. However, it was challenging to make an assumption when we designed this study in 2019. At that time, ICI only showed efficacy in some studies, such as ATTRACTION-02 (Kang, *Lancet*, 2017) and KEYNOTE-059 (Fuchs, *JAMA Oncol*, 2018) in the late-line setting, but failed in the other studies (e.g., JAVELIN 300; Bang, *Ann Oncol*, 2018). In the second-line setting, ICI also failed (e.g., KEYNOTE-061; Shitara, *Lancet*, 2018). Moreover, the trial outcomes in the first-line setting were unavailable (like Checkmate 649 and KEYNOTE-062). Most importantly, there were no reported trials on neoadjuvant ICI-based treatment in gastric cancer, so we were unable to estimate a pathological response rate, which is commonly used to evaluate the efficacy of neoadjuvant treatment for sample size calculation in phase II trials.

Therefore, we set a sample size for an exploratory study following similar trials in lung cancer and melanoma (Forde, *NEJM*, 2018 and Amaria, *Nature Medicine*, 2018). To our knowledge, recent similar studies also recruited about 20 - 30 subjects (Chalabi, *Nature Medicine*, 2020 and ASCO abstracts DOI:

*J Clin Oncol.*2021.39.15_suppl.4040, 4046, 4026). After 2021, some neoadjuvant ICI studies in gastric cancer were reported (e.g., 2021 ASCO abstracts DOI: *J Clin Oncol.*2021.39.15_suppl.4040, 4026, and 4046). We have summarized these meeting reports as a meta-analysis (2022 ASCO-GI abstract DOI: *J Clin Oncol.*2022.40.4_suppl.291) and obtained our own results from this clinical trial. These results provide references for power calculations and sample size determination in our future study. We will use the available outcomes to calculate sample sizes with the help of statisticians in our future trials.

4. A standardized pathologic assessment system such as Becker or Mandard should be used for path assessment. What system was used here? This is important to increase comparability to other trials which the authors do in the discussion.

Response: Thank you for raising this valuable question. Considering the lack of consensus on the TRG system after immunotherapy, we learned from the pilot studies on lung cancer (Forde, *NEJM*, 2018) and melanoma (Amaria, *Nature Medicine*, 2018). These studies estimated percentages of vital tumor cells in tumor beds and defined <10% residual tumor cells as MPR. This is often used as a key endpoint in subsequent trials with neoadjuvant ICI therapy, e.g., in early-stage colon and lung cancers (Chalabi, *Nature Medicine*, 2020; Forde, *NEJM*, 2022; and trial names: IMpower-030, AEGEAN, and BGB-A317-315).

We agree that a scoring system for comparison with the other trials should be used. Therefore, we re-assessed tumor regression with the Becker system (**Methods, line 445 - 448**), which was also used in the FLOT4-AIO trial. The grading according to the Becker system was described in the **Results (lines 114 - 116)** and **Table 2**. Ours and Becker grading systems can be converted to each other. Our predefined CPR is equivalent to Becker TRG1a, MPR is equivalent to TRG1a+1b, and PPR is equivalent to TRG1a+1b+2.

Results:

1. Table 1: It seems that PD-L1 CPS is shown but would be nice for this to be explicitly stated.

Response: Thank you for this suggestion. We categorized the CPS score into <1, 1 - 5, 5 - 10, and ≥10 in **Table 1**. In addition, the detailed CPS scores of individual patients have been listed in **Supplementary Table S1**.

2. Line 83-84: additional detail on how many doses and what neoadjuvant treatments exactly were administered would be beneficial. It would be helpful to depict these details graphically or perhaps as a supplementary table. These details are especially important given the small sample size and therefore increased importance of each patient's therapeutic context.

Response: Thank you so much for this suggestion. As you've suggested, we listed each patient's baseline characteristics (age, gender, ECOG, T stage, and PD-L1 CPS score) and treatment details (medicines and treatment cycles) in **Supplementary Table S1** and described them in the **Results (lines 84 – 85)**. We believe showing the treatment details would be essential for understanding the trial results.

3. With such a small sample size, inclusion of 4 dMMR patients risks a dominant driver effect in efficacy from this minority fraction of patients. It would be helpful to have analyses presented with dMMR/MSI-H patients excluded so as to better understand the magnitude of benefit in pMMR/MSI-low patients. For example, in Supp Fig 3, are there still significant pre/post-treatment changes in TNB and TMB if P01 is excluded?

Response: Thank you for this comment, which we think requires further detailed analysis. In this study, the patients were recruited in chronological order, with no preference for any pathological type. In total, four MSI-H patients were recruited, one of whom refused surgery. The other three MSI-H patients all achieved R0 resection and MPR. As you suggested, we re-analyzed the WES and transcriptome data by excluding dMMR/MSI-H patients (3 and 2, respectively). No MSI-H patients were included in the TCR analysis, so there was no revision for TCR data. The amendments are as follows:

Fig. 1B and 1C: MSI-H patients were marked by “*”. The figure legends were also changed accordingly.

Fig. 3C: TMB comparison between MPR and non-MPR were re-analyzed after MSI-H samples were excluded.

Fig. 3D-E: TMB changes were analyzed after excluding MSI-H samples.

Supplementary Fig. S5B: Exome-wide dN/dS were analyzed without MSI-H samples.

Fig. 3G: MSI-H patients were marked by “*”.

Fig. 4D, 4E, 4G, and 4H: The corresponding analyses on MSS-only patients were added in **Supplementary Fig. S9, S10, S11, and S12**.

Fig. 5A: MSI-H patients were marked by “*”.

By excluding MSI-H samples, we found the MSS patients also showed similar differences in some biomarkers but might lose significance due to the limited sample size. For example, the MPR tumors had a higher baseline TMB than non-MPR tumors (16.78 vs. 5.27 Muts/Mb), but without significance ($P = 0.067$, **Fig. 3C**). The TMB drop from pre- to post-treatment samples still remained significant in all MSS patients and non-MPR patients after MSI-H patients were excluded (**Fig. 3D**). In transcriptome and immune cell estimation analysis, the MSS MPR samples had similar levels/trends with the MSI-H patients in most analyses. Unfortunately, statistical tests cannot be performed because only 1 MSS PPR+ sample was available (**Fig. S9, S10, S11, S12**). We infer that most of these data supported our previous conclusion.

To our knowledge, although MSI-H tumors were generally regarded to respond well to ICI, a good response was not guaranteed. Primary resistance of MSI-H tumors to immunotherapy has recently attracted wide attention. The ORR rates for ICI monotherapy in advanced MSI-H gastric cancer were only 57%, 47%, 57% in Keynote-059, -061, and -062 trials (Chao, *J Clin Oncol*.2020.38.4_suppl.430). Even ICI plus chemotherapy in checkmate-649 only obtained an ORR rate of 55%, and about 30% of patients deceased in one year in this subgroup (Janjigian et al., 2021 ESMO, LBA 7). Before our study, few articles reported outcomes of neoadjuvant ICI-based therapy in MSI-H gastric cancer, so we think the results of MSI-H gastric cancer may also have some value.

4. Since pathologic response is a primary outcome, movement of Table S1 into the main manuscript should be considered, especially pathological complete response rate as this outcome helps interpretation/comparison to other landmark trials such as FLOT4-AIO.

Response: We agree with you. We have moved **Supplementary Table S1** to **Table 2** and added grading according to the Becker TRG system in this table.

5. Figure 2A: Tumor fraction/purity should be reported along with TMB so as to increase clarity. As presented, the data risks suggesting that TMB in the same tumor cells decreases after treatment which is incorrect.

Response: Thank you for pointing this out. We agree that the tumor purity might influence TMB. To address this possibility, we estimated the tumor purity of all samples from the WES data computationally (**Method, lines 490 - 491**). The results were added in **Fig. 3A**.

Considering the association between TMB and purity, we performed a simulation to balance the purity of pre- and post-treatment samples by blending tumor and control samples (**Method, lines 491 - 498**). After

the simulation, the purities, coverages, and depth were approximately equal and comparable. In this case, we found a decrease in TMB in all MSS or non-MPR patients as before (**Supplementary Fig. S5B**). The results were described in the **Result (lines 168 - 171)**.

6. The findings in Figure 2F for pre/post microenvironment changes in cellular composition is interesting and well demonstrated. A side comparison in a patient with non-MPR would strengthen this argument.

Response: Thank you for this advice. We agree with you and performed multiplex IF on two patients with non-MPR (P10 and P20). We also found infiltration of immune cells in these two tumors post-treatment. In P10, there were increased CD8⁺ cells in the post-treatment specimen. In P20, there was an increase in CD8⁺ cells and macrophages in the post-treatment tumor, while many T_{reg} cells remained. By contrast, the MPR patient, P18, had CD8⁺ cell influx to a greater extent. The results were shown in **Supplementary Fig. S2** and described in **Results, lines 134 - 137**.

7. I would recommend softening the statement regarding co-localization of macrophages and tumor cells equating to phagocytosis as an anti-tumor mechanism, further mechanistic work is needed to definitively state this.

Response: Thank you for this comment. We agree that the co-localization does not definitively indicate phagocytosis. Therefore, we deleted the assertion that co-localization suggests phagocytosis and only stated the phenomenon of co-localization in the revised manuscript (**Results, lines 133 - 134**).

8. Regarding results shown in 2G and 2H. Why do the authors think we see these trends? Do they believe this is from immunoediting?

Response: Thank you for raising this valuable and important question. To make it as clear as possible, we studied the literature, discussed it with several immunologists and bioinformatics experts, and conducted some tests.

First, we used a simulation method to balance the tumor purities to eliminate their influence. After the purity adjustment, we still observed a drop in TMB in responders, non-responders, or MSS populations. Then we used the relative rates of non-synonymous and synonymous mutations (dN/dS ratio) to evaluate clonal selection (Minsuk Kwon, *Cancer discovery*, 2021). The dN/dS ratio was estimated by the package “*dNdScv*” (Martincorena, *Cell*, 2017) using trinucleotide mutational signatures, sequence composition, and variable mutation rates across a set of homologous protein-coding genes. By this algorithm, we found the dN/dS significantly dropped after treatment in patients with good response (PPR+), indicating a

negative selection (**Supplementary Fig. 5B and Results, lines 171 - 174**). Moreover, the PyClone results showed subclonal changes in these patients (**Fig. 3I**). The above evidence indicated that the change of TMB and TNB (**Fig. 2G**) was a negative selection process. Although the driving agent (chemotherapy vs. ICI vs. apatinib) remains unclear due to this single-arm study with combined treatment, with the coincident T cell expansion (**Fig. 2H**), we infer that this selection was, at least partially, driven by immunotherapy and, thus, was a result of immunoediting.

9. Supp Fig S2-can the authors define RCCEP? Could include in legend.

Response: Thank you for this suggestion. RCCEP means reactive cutaneous capillary endothelial proliferation and is a common adverse event caused by camrelizumab. We have defined it in the legend of **Supplementary Fig. S2**.

10. Could the authors comment on how these data inform optimal time to surgery? This is important as we begin to think about the temporal dynamics of an optimal anti-tumor immune response in the neoadjuvant setting. It is possible that with a mean of 16 days and a tight range (14-19 with one exception), there was not adequate time to capture a full materialized immune response. Similarly, there is a possibility that a longer duration of neoadjuvant therapy would improve outcomes. In fact in clinical T4 patients one could argue for a more extended duration of chemotherapy such as in the FLOT4 trial. This warrants comment.

Response: Thanks for this question, which we think is vital to neoadjuvant therapies. As you mentioned, treatment duration and interval are important to neoadjuvant therapy. Based on our years of observation in clinical trials and in clinical practice, and the characteristics and mechanisms of immunotherapy, we now realize that a longer interval before surgery and more prolonged treatment duration might improve the outcomes of neoadjuvant immunotherapy. As in the recent DANTE trial (2022 ASCO Abstract 4003), they used four cycles of neoadjuvant atezolizumab combined with FLOT. Unfortunately, in 2019, we did not have many pioneers and mainly followed the design of Checkmate-159, which achieved an excellent outcome by two cycles of neoadjuvant ICI treatment in operable NSCLC (Forde, *NEJM*, 2018). We also did not know whether this neoadjuvant strategy could benefit most patients or it might diminish the opportunity for eradication surgery at that time. Taking all of these factors into consideration, we designed the protocol: patients would receive efficacy evaluation every 2 cycles of treatment and would undergo surgery once the tumor is resectable, especially for the T4bN+ ones with conversion therapy. In practice, 20 patients received 2 cycles of treatment, 3 received 4, and 2 received 6 cycles (**Supplementary Table S1**).

Due to the trial design, our study may not be able to instruct the optimal time for surgery. Based on the present results, our design with at least two cycles of treatment and >14-day interval before surgery is feasible and safe. Further studies with different treatment cycles and time to surgery should be conducted to determine an optimal treatment duration and time to surgery, as in the recent NeoSCORE trial for NSCLC (2022 ASCO Abstract 8500). To make up for this question, we have commented on the above issues and mentioned the need for further exploration in the revised manuscript (**Discussion, lines 403 - 407**).

11. Supp Fig 3: I don't understand 3c that shows no significant difference in pre/post change of TMB, TNB and MSI yet the statement in the manuscript is that there is a significant change (line148-149)

Response: Thank you for raising this question. We apologize for misusing "significantly". We corrected the statement in the manuscript that we observed a decrease but without statistical significance (**Results, lines 163 - 165**).

12. Line 179-181: Reads a bit awkward as currently phrased. Suggest breaking into two sentences for increased clarity.

Response: Thank you so much for this advice. During the revision, we found the 293 genes may not be valuable in predicting responses. After our discussion and consideration, we decided to delete the description of the 293 genes in the manuscript (**Results, line 204 - 208; Discussion, lines 352 - 353**).

13. Line 206-209: The statement that neoadjuvant therapy promoted cytolytic and CD8+ effector signatures in all patients (4H and Supp fig 8) should be modified to clearly state that these differences were not statistically significant (presumably due to being underpowered)

Response: Thank you for pointing out this problem. The sample numbers, especially the patients with PPR+, were indeed limited. For correction, we stated that these changes were not statistically significant, probably due to low power (**Results, lines 237 - 238**). We believe that this improves the manuscript's clarity and accuracy.

14. Figure 5: Limited power should be clearly stated in limitations

Response: Thank you for this comment. Some of the gastroscopy specimens were too small to prepare libraries of transcriptome sequencing, resulting in a small sample size for analysis in **Fig. 5**. Paring pre- and post-treatment samples further added difficulties. Therefore, we emphasized that the immune cell

analysis was limited by sample sizes (**Discussion, line 378 - 380**). In addition, we checked other parts with similar problems in the manuscript. These were also discussed as a limitation (**Discussion, lines 394 - 397**).

15. Figure 5D: If available, it would be nice to see this responder/non-responder anecdotal comparison but for the aggregate cohort instead of selected single patients. This could be included in the supplement.

Response: Thanks for your advice. Initially, we used this to describe the local immune status of two individual patients (similar to the multiplex immunofluorescence). We agree that it is better to compare the aggregate cohort. Therefore, the Immunophenoscore analysis of all patients was displayed in **Fig.5E and Supplementary Fig. S14**. We found EC and MHC increased in both PPR+ patients, with stable SC and CP. In non-PPR+ patients, a group of tumors had increased EC and MHC along with increased SC and CP, while the other group had no obvious TME changes. These results were described and discussed in **Results, lines 249 - 258**.

16. Figure 5A could be depicted more clearly. For example, left of patient ID #, PPR+ vs non-PPR+ status could be shown.

Response: Thank you for this suggestion. We have revised this Figure as you suggested, by adding the patient ID # and pathological response status (PPR+ vs. non-PPR+). We believe this makes the description clearer (**Fig. 5A**).

17. Figure 6: Is anything known in the literature about the specific VDJ sequences observed?

Response: Thank you for raising this question. Through literature reviews, we found some studies that reported specific V and J usages in cancer. For example, TRBV24-1 and 25-1 and TRBJ2-5 and 2-7 were highly used in prostate cancer compared to para-cancer samples (Liu, *Oncol Lett.* 2018). However, few studies reported the dynamic changes of V and J usages during treatment, and we did not find any reports of the dynamic changes TRBV 20-1 and TRBJ 2-5 during therapy.

18. Figure 6: Increased clarity would be appreciated regarding whether the entire analysis is from peripheral T cells or if TCR analysis using tumor tissue was used as well.

Response: Thank you for your feedback. All of the TCR analysis was from peripheral blood, and no tumor-infiltrating T cells were sequenced. It has been reported that the neoantigen-specific T lymphocytes were identified in peripheral blood and overlapped with those in tumor tissues (Gros, *Nature Medicine*,

2016), so we suppose that the peripheral TCR sequencing may represent the tumor-infiltrating cells to some extent. For this issue, we made revisions to clarify that no tumor-infiltrating T cells were analyzed and that TCR repertoire overlapped between peripheral and tumor-infiltrating T cells in the **Discussion (lines 391 - 393)**. Besides, we mentioned that the TCR sequencing is performed from peripheral T cells in the **Results (line 261)** and **Methods (line 534)**.

Discussion:

1. A limitation of this small (but data dense) study is that characterization of the relative contribution of each therapeutic component ie: chemotherapy versus checkpoint inhibitor versus anti-angiogenic agent is lacking.

Response: We agree with your comment. It is a limitation that this single-arm study, designed to explore the primary efficacy of a combined regimen, cannot dissect the relative contributions of each treatment component. Under your advice, we added this limitation in the revised Discussion part to point out this limitation (**Discussion, lines 401 - 403**). We expect to address these issues in large randomized controlled trials in the future.

2. The multiplex immunofluorescence work was quite limited and descriptive in nature. If additional data is available, it would have beneficial to include with quantification of changes in cellular composition such that comparison could be made with the results of cellular deconvolution from bulk transcriptome data. Further, beyond the suggestion that macrophages co-localize with tumor cells, there is no discussion of tumor and tumor microenvironment dynamics in the spatial dimension and additional multiplex IF data could potentially explore this. An representative study (albeit focused on colorectal cancer) is Bortolomeazzi et al, Gastroenterology 2021; 161:1179-1193

Response: This is an extraordinarily valuable comment. We carefully read the paper you recommended and agree that quantifying cellular composition changes by multiplex immunofluorescence would be decisive for exploring the TME evolution during the treatment. Unfortunately, our work is extremely limited by sample availability, because many pre-treatment tumor tissues were obtained and deposited in district hospitals. Also, for paired tissues in our institution, many of them are too small to be sectioned. Despite this, we tried our best to get two pairs of specimens from patients with non-MPR (P10 and P20) and performed multiplex immunofluorescence as suggested. We found CD8⁺ cell infiltrations after treatment in these two patients, yet to a lesser extent than in the MPR sample. We also found many T_{reg} cells in both pre- and post-treatment samples from P20. The results were shown in **Supplementary Fig. S2** and **Result lines 134 - 137**, and **Discussion lines 369 - 371 and lines 373 – 374**.

Though not quantitative and satisfactory, we hope our additional effort provides a preliminary and descriptive comparison between MPR and non-MPR patients, as well as the heterogeneity among the non-MPR samples. We plan to pre-design multiplex immunofluorescence tests and preserve slides in our future trials for comprehensive analysis of the TME changes during immunotherapy. Thank you again for these valuable comments.

3. Line 260-261: This statement should be softened because pathologic complete response is comparable to phase III data from FLOT4.

Response: Thanks for this comment. Indeed, the FLOT4-AIO trial showed a numerically similar pCR rate, and we should draw the conclusion objectively instead of overstating it. Notably, the patients in our cohort (all cT4N+, 56% were cT4bN+) are more advanced than those in FLOT4 (cT4 patients < 10%), being predisposed to have a worse prognosis. In addition, in FLOT4, pCR mainly occurs in the intestinal type of Lauren's classification, with only 3% in the diffused type (Al-Batran, *Lancet Oncol*, 2016). By contrast, in our study, all patients with pCR were diffused types and the pCR rate in diffused type was 30% (3/10).

It's interesting that the DANTE trial (2022 ASCO Abstract 4003) also reported a nearly similar TRG1a rate between the atezolizumab plus FLOT versus FLOT only subgroup (25% vs. 24%) in the ITT population, while atezolizumab plus FLOT achieved much better TRG1a rates over FLOT chemotherapy along with the increasing level of CPS expression (46% vs. 24% in CPS \geq 10 patients). Therefore, we think that adding ICI might offer benefit beyond chemotherapy to some extent, at least for certain subgroups.

As you suggested, we have softened the statement by deleting the word "superior" in the revised manuscript (**Discussion, lines 295 - 296**). We also read through the entire manuscript and softened similar statements that adding neoadjuvant ICI is better (e.g., **Conclusion lines 408 - 411**). In addition, we briefly discussed the differences in baseline characters as described above (**Discussion, lines 303 - 309**). We believe these changes make the conclusion more accurate.

4. A limitation of this study is relatively short follow up (~15 months). It remains unclear if benefit in more longitudinal outcomes DFS, PFS, OS is seen which is highly relevant in the context of increase anti-tumor immunity which would theoretically decrease distant metastatic recurrences.

Response: This is quite a good idea that provides an opportunity to update our research data. The last follow-up was by July 2021 for the manuscript, but they were still under regular follow-up. To report outcomes with more extended observation, we updated the follow-up by June 2022 to make a median

follow-up time of 24.7 (IQR 20.9-31.8) months. We added the updated data to the revised manuscript accordingly (**Fig. 1B and D; Results, lines 86 - 88**).

5. Line 270-272: This is an overstatement given that multiplex IF was only shown for 1 patient.

Response: Thank you for noting this overstatement. Due to limited sample availability, we tried our best to add 2 pairs of samples for multiplex IF staining and collected new data. As suggested, we softened the description of the multiplex IF results and emphasized that this was from several individual patients (**Discussion, line 369 - 371**).

6. The authors should be commended for including a high risk population (cT4N+) in their study.

Response: Thanks for your encouragement.

7. The discussion should include commentary on how the putative biomarkers explored in this study have the potential to outperform PD-L1 for predicting response to checkpoint blockade. Have the authors offered data to support a superior marker? For example, does TCR and WES data add any predictive power beyond PD-L1 status in identifying pathologic responders. This is an important question for the field and it is possible that PD-L1 expression is the end result of the favorable features (high TMB, etc) that the authors identify here.

Response: We think this is also important advice. In the previous submission, we only analyzed the predictive values of individual biomarkers but failed to compare them, especially in the case of good performance of PD-L1. Indeed, three of five patients with MPR were PD-L1 positive. Following your guidance, we compared the putative biomarkers with PD-L1 and summarized the description in the Discussion. As expected, MSI-H, TMB-H, and TNB-H showed a strong association with MPR, with $P = 0.0088$, 0.0049 , and 0.0088 (**Supplementary Table S10**). By contrast, *RREB1* and *SSPO* mutations showed comparable or even better association with MPR ($P = 0.0010$ and 0.0049). *SSPO* mutation, *RREB1* mutation, and TMB were present in PD-L1-negative MPR patients. In particular, the combination of PD-L1 and *RREB1* mutation was present in 100% MPR patients and 0% non-MPR patients. In predicting partial pathological response+ (PPR+, tumor cells < 30%), the dynamic TCR data, CE score, showed better values. Based on the comparison, we believe the above biomarkers showed preliminary values alone or in combination with PD-L1. To explain these, we drew a table (**Supplementary Table S10**) to show the comparison and discussed in the revised manuscript (**Discussion line 340 - 347**).

8. The overall discussion needs to be tempered to avoid overstating the impact of the data. There is for example no evidence that bringing immunotherapy into the neoadjuvant setting in gastric cancer is superior to chemotherapy.

Response: Thank you for this suggestion. Indeed, some neoadjuvant trials, such as FLOT4-AIO, showed a numerically similar pCR rate. However, we believe this exploratory trial adds additional value. As we mentioned above, the baseline characters in our trial were much more advanced (all were T4a/bN+, including 56% T4bN+, and 64% were Lauren' diffused types) than in the other trials, being predisposed to have a worse prognosis. Moreover, our treatment showed excellent efficacy in MSI-H patients, in that ICI or ICI plus chemotherapy only has less than 60% ORR in MSI-H advanced gastric cancer (Chao, *J Clin Oncol.*2020.38.4_suppl.430; Janjigian et al., 2021 ESMO, LBA 7), possibly supporting the usage of immunotherapy in the earlier stages. Even in MSS patients, the combined treatment remodeled the TME to a "hot" status, so we suppose this may improve outcomes if the treatment duration is prolonged. In addition, candidate biomarkers discovered in this trial might be utilized to add to the predictive values to PD-L1 and MSI, especially in the neoadjuvant setting. Considering the result of recent DANTE trial (2022 ASCO Abstract 4003), we think that adding ICI might offer benefit beyond chemotherapy to some extent, at least for certain subgroups.

As suggested, we have read through the entire manuscript and softened statements that adding neoadjuvant ICI is better (e.g., **Discussion lines 295 - 296; Conclusion lines 408 - 411**). In addition, we compared our results with the other trials (**Discussion, line 303 - 306**). We believe these changes make the discussion more objective.

Minor Points:

Abstract:

1. Line 30: suggest swapping unsatisfactory to "limited" or "poor"

Response: Thank you for this suggestion. We have replaced "unsatisfactory" with "poor" (**Abstract, line 29**).

2. Line 30: There is a typo in "antiangiogenic"

Response: Thank you for noting this typo. We apologize for this misspelling. We have corrected them throughout the manuscript (**lines 63, 70, 73, 291, and 318**).

3. Line 30: suggest swapping "effect" for activity

Response: Thank you for this suggestion. We have made this change in the manuscript (**Abstract, line 30**).

4. Line 33-34: The treatment arms were not introduced clearly yet and thus it is unclear to which treatment arm the path response rates pertain to.

Response: Thank you for this suggestion. Following your advice, we have added the treatment components in the revised manuscript (**Abstract, lines 31 - 32; Methods, lines 425 - 426; Supplementary Fig. S1**).

5. Line 35: suggest removing 's' in "tumor mutational burdens"

Response: Thank you for this suggestion. We have made the change in the revised manuscript (**line 35**).

6. Line 39: suggest "...during neoadjuvant immunotherapy."

Response: Thank you for this suggestion. We have changed this sentence as you suggested (**line 38 - 39**).

7. Line 41: suggest deleting "efficacy-related"

Response: Thank you for your suggestion. We have deleted these words (**Abstract, line 41**).

Introduction:

1. Line 62-63: Suggest swapping "was an attractive and critical question" for "...remains incompletely explored."

Response: Thank you for your instruction. We have made the suggested change (**lines 61 - 62**).

2. Line 64: typo in "anti-angiogenetic" (the manuscript should be checked for this typo as it occurs in several additional places beyond line 64).

Response: Thank you for your suggestion. We have checked the entire manuscript, figure legends, and tables and found several such typos. All of them were corrected (**lines 63, 70, 73, 291, and 318**).

3. Line 64: Inclusion of a hyphen ie: Anti-angiogenic versus antiangiogenic should be decided upon and the manuscript should be uniform

Response: Thanks for your comment. We have checked the manuscript and made the suggested changes (**lines 63, 70, 73, 291, and 318**).

4. Line 75: typo in "whole-exon", should be "whole exome sequencing"

Response: Thank you for this suggestion. We apologize for this misspelling. We have corrected it in the manuscript (**line 75**).

We thank you again for carefully reviewing our manuscript and offering critical and constructive suggestions, which greatly improve our manuscript.

Reviewer #2 (Remarks to the Author): with expertise in cancer genomics

Li et al. have evaluated the efficacy of ICI and antiangiogenic agents in late-stage gastric cancer in the neoadjuvant setting, and further investigated the genomic, transcriptomic and immune-related features associated with pathogenic response. Overall, the topic and approach of the manuscript is interesting and should be of interest to Nature Communications readers. However, I have several major concerns, especially on the WES analysis and the reported results on SSPO mutations.

Response: Dear professor, thank you for your careful review and detailed comments, which have strengthened our study. We have carefully considered every comment, performed further experiments and analysis, and made amendments to the manuscript. Notably, we used three independent mutation callers (Mutect2, VarScan, and Strelka2) to recall the mutations for more reliable results. In addition, we used a simulation method to address the purity imbalance issue. All of the results involving mutations were updated. Moreover, the SSPO mutation issues are explained in this response letter as well as in the manuscript. The responses and revisions are detailed in the following pages. For your convenience to review, we uploaded two manuscripts. One is with track changes, its line numbers being referred to by the response letter. A clean version was in the supplementary files, with the revisions colored in red.

- 1. I could not reproduce the reported association between SSPO (SSPOP) mutations and overall survival in the TCGA STAD cohort. SSPO is mutated in only 3/478 (0.6%) patients in the STAD cohort based on data from cBioPortal, and there is no association (P = 0.3) between SSPO mutations and overall survival:**
https://www.cbioportal.org/results/oncprint?cancer_study_list=stad_tcga&Z_SCORE_THRESHOLD=2.0&RPPA_SCORE_THRESHOLD=2.0&profileFilter=mutations&case_set_id=stad_tcga_sequenced&gene_list=SSPOP&geneset_list=&tab_index=tab_visualize&Action=Submit&comparison_subtab=survival

Response: Thank you for your comment. We performed the survival analysis using *SSPO* non-silent mutations in the TCGA STAD cohort and obtained the same result that you did. Yet, *SSPO* is a pseudogene, as you mentioned. Given that a pseudogene works based on its RNA sequences (e.g., competing miRNA or RNA-binding protein; Roberts, Pharmacogenomics, 2013), we included both silent and non-silent mutations for the survival analysis. The result was a significant association with overall survival.

We have stated that we used both silent and non-silent mutations for survival analysis (**Results, line 177**), and the predictive value of non-silent mutations in TCGA-STAD was insignificant (**Results, lines 178 - 179**).

2. I noticed that the mutation calling was performed with Varscan using a VAF cutoff of 0.02 (2%). The authors did not specify the sequencing coverage of the WES samples. Assuming 100x coverage, a position with VAF=0.02 translates to only 2 supporting reads. At such a low VAF cutoff, the mutations called would have very low precision with a large number of false positives. Furthermore, recent benchmarking papers found Varscan to perform poorly compared to other mutation callers (Wang et al. SomaticCombiner: improving the performance of somatic variant calling based on evaluation tests and a consensus approach. *Sci Rep* 10, 12898 (2020). <https://doi.org/10.1038/s41598-020-69772-8>; Huang et al., SMuRF: portable and accurate ensemble prediction of somatic mutations, *Bioinformatics*, Volume 35, Issue 17, 1 September 2019, Pages 3157–3159, <https://doi.org/10.1093/bioinformatics/btz018>). I would recommend that the authors use an additional best-in-class mutation caller (such as Mutect2 or Strelka2) to produce high confidence mutation calls (e.g. taking the intersection of Mutect2/Strelka2/Varscan).

Response: Thank you for this suggestion. In this study, the average sequencing coverage was 332x, so the VAF = 0.02 could translate into 6 supporting reads. The quality control information was added in **Supplementary Table S11**. We agree that one caller might not be enough for high-confidence somatic variant calling. Per your suggestion, we called mutations by Mutect2, Varscan, and Strelka2. Overlaps

among ≥ 2 callers were considered mutations (described in the **Methods, lines 466 - 468**). As a result, we obtained 14,316 non-synonymous mutations and 6,037 synonymous mutations. The revised results do not significantly change the conclusion, including frequently mutated genes, TMB, and other analyses. However, after the revision, *RREB1* mutation showed a better predictive value than *SSPO*.

Based on the new results, the figures on mutations, TMB, TNB, and PyClone were re-analyzed, including **Figures 2G, 3A, 3B - E, 3G - I, S5 - 7, and Table S2 - 4**.

3. The authors identified the *SSPO* gene to be frequently mutated in MPR patients, but not in non-MPR patients. What is the functional role of the gene/protein (it is annotated as a transcribed pseudogene by Ensembl)? Where are the mutations found in the *SSPO* protein?

Response: Thank you for these interesting questions. In animals (e.g., rats, bovine, and zebrafish), *SSPO* was reported to regulate protein multimerization, low-density lipoprotein (LDL) recognition, transforming growth factor β (TGF- β) activation, and neuroinflammation. However, it was reported to be a pseudogene in Entrez Gene (<https://www.ncbi.nlm.nih.gov/gene/23145>) and HGNC (https://www.genenames.org/data/gene-symbol-report/#!/hgnc_id/21998) in humans. To date, no study has reported its role in humans or cancer. One study reported that it could not be detected in apes and humans (Rodríguez, Int Rev Cytol, 1992). Our recent study found that *SSPO* is also frequently mutated in gastric cancer peritoneal metastasis (unpublished data).

The above information has been added to the manuscript (**Discussion, line 347**). This manuscript only focused on the association between gene mutations and pathological responses. We are eager to investigate the underlying mechanisms in future studies.

By recalling the mutations, *SSPO* mutation occurred in 4/5 MPR samples and in 1/14 patients at baseline. The mutations include 9 missense mutations, 2 frameshift mutations, and 1 nonsense mutation. There were also recurrent mutations at 4108 and 4834 and around them among the 5 patients. The mutation distribution has been displayed in **Supplementary Fig. S5C**.

4. Also, the figure legend for frameshift mutations seems to be wrong in Figure 3A.

Response: Thank you for noting the problem in **Fig. 3A**. We apologize for this mistake and re-drew the landscape plot. Recalling mutations makes the frequencies of mutations different from the previous ones. The revised plot has been updated in **Fig. 3A**.

5. The authors found a general decrease in TMB post treatment, and concluded that neoadjuvant therapy eliminated sensitive clones. Similarly, the authors report pathological regressions were correlated with 'loss' of mutations and subclones (Fig. 3E/F). However, the authors have omitted to explore whether the decrease in the number of mutations detected could be explained by reduced tumor purity (viable cancer cells) post treatment. Tumor purity can be estimated with NGS/WES data. The authors should rule out that the observed 'loss' of mutations / clones / decreased TMB is not just due to a drop in tumor purity in tumors with pathological response.

Response: Thank you for pointing out this important issue. We agree that the tumor purity might influence TMB. Under your instruction, we estimated the tumor purity of all samples from the WES data computationally (**Methods, lines 490 - 491**). The results have been added to **Fig. 3A**. Furthermore, we performed a simulation to balance the purity of pre- and post-treatment samples by blending tumor and control and then recalled the mutations (**Methods, line 491 - 498**) to determine whether the decreased TMB was only due to the drop in purity. After the simulation, the purities, coverages, and depth were approximately equal and comparable. We found a decrease in TMB in all MSS or non-MPR patients, as before (**Supplementary Fig. S5B**). The results have been described in the **Results (lines 168 - 171)**. Based on these, we suspect that the decreased TMB was not solely due to purity.

6. The authors defined clonal genes as "For all patients with both pre- and post-treatment samples, if the change of mutation VAF was consistent with that of its clone's cellular prevalence, the host gene of the mutation was identified as a clonal gene". This definition can

be confusing as people generally regard clonal mutations as mutations that are present in all subclones of the tumor.

Response: Thank you for your comment. We agree with you that this part was confusing. Given that the 293 genes may not be valuable in predicting responses (as noted in Comment #8 and its response), we have deleted this part in the manuscript (**Results, line 204 - 208; Discussion, lines 352 - 353**).

7. The authors found that CD274 (PD-L1) expression decreased in post treatment samples (Figure 4E). On the other hand, they also found the number of PD-L1+ patients increased post treatment (Figure 2C). Can the authors elaborate/discuss on these seeming conflicting results?

Response: Thanks for your careful review and for raising this question. With the Wilcoxon rank-sum test, the decrease of PD-L1 mRNA was not significant ($P = 0.057$), but there was a downward trend after treatment. In our opinion, there may be two reasons for this inconsistency: First, there was not a good overlap between the patients with IHC and RNA-seq analysis for partial pathological response+ (PPR+) patients. Six out of fifteen patients had elevated PD-L1 expression by IHC (**Fig. 2C**), but only 1 of them was involved in the RNA-sequencing analysis for PPR+ patients (**Fig. 4E**). When analyzing PD-L1 mRNA in non-PPR+ patients, 5 increased and 4 decreased (**Supplementary Fig. 10C**). Second, PD-L1 expression was regulated on many levels, including epigenetics, transcription, and post-translation (Sun, *Immunity*, 2018; Ju, *Am J Cancer Res.* 2020). Post-translational regulation is one of the important regulation mechanisms and may cause an inconsistency between protein and mRNA levels. Many post-translational regulators, such as CMTM6, CMTM4, CSN5, GSK3B, and B3GNT3, have been discovered (Sun, *Immunity*, 2018). These potential reasons have also been discussed in the manuscript (**lines 229 - 231**).

8. P. 9 I. 181: "These genes may be candidate biomarkers for predicting responses to neoadjuvant therapy." The authors identify 293 clonal genes that either 'expand' or 'contract' in VAF upon treatment. However, it is not clear how these genes are useful as candidate biomarkers for prediction of response at baseline (before treatment).

Response: Thank you so much for this advice, which another reviewer also pointed out. After our discussion, we totally agree that the 293 genes may not be valuable in predicting responses. Therefore, we deleted the content reporting the 293 genes in the manuscript (**Results, lines 204 - 208; Discussion, lines 352 - 353**).

Minor points:

- 1. Previous studies suggested that EBV positive gastric cancer tend to respond favorably to ICI. EBV typically comprise of around 10% of gastric cancer cases. It would be interesting to know if EBV patients also respond well in the neoadjuvant setting. Are any of the patients in this study EBV positive? Is so, could the authors annotate EBV patients in Figure 2?**

Response: Thank you for your suggestion and questions. EBV positive gastric cancer responds well to ICI, which is regarded as a good predictive biomarker. In our research, there was no EBV positive case who received the ISH test of EBER, so we didn't mention it in the manuscript.

- 2. P-values should be added to Figure 1D**

Response: Thanks for your recommendation. We have added P-values to **Fig. 1D**.

- 3. Figure legend is missing for Figure 2C.**

Response: Thank you for noticing our mistake. We apologize for this and have added the legend for **Fig. 2C (Figure legends, lines 891 - 892)**.

- 4. For Figure 4C and 4F, what do the colors of the heatmaps represent? Z-score? The figure legend should be made more descriptive.**

Response: Thank you for the feedback. The heatmap colors indicate $\log_2(\text{TPM} + 1)$ and the signature score in **Figs. 4C and 4F**. We have added the descriptions to the figure legends.

5. The boxplots in Figure 5B show large fold change but all the p-values are not significant. Can the authors confirm the p-values are correct and perhaps show the data points on the plots?

Response: Thank you so much for this feedback. We have re-checked the data and showed all data points and actual *P* values in the figures (**Fig. 5C** and **5D** in the revised edition). The statistical insignificance might be limited by sample numbers and non-normal distribution (tested by Wilcoxon signed-rank test).

6. Why fit a regression line through the two sets of points in Figure 3G? What's the purpose of figure 3G?

Response: Thank you for these questions. The **previous Fig. 3G** was drawn to show the distribution of VAF of the 293 genes; one group was high in the pre-treatment sample, and another was high in the post-treatment samples. After discussion with several statisticians, we realized that the regression was not meaningful, so deleting the regression line is appropriate. As we mentioned above, 293 clonal genes do not have a predictive value, so the related content was removed. As a result, the figure was also deleted.

Again, we thank you very much for carefully reviewing our manuscript and providing critical and constructive suggestions, which have greatly improved our manuscript.

Reviewer #3 (Remarks to the Author): with expertise in gastric cancer, clinical

This manuscript describes the results of a phase II trial in which patients with resectable gastroesophageal cancer were treated with chemotherapy, a PD-1 inhibitor and an antiangiogenic TKI. All patients had locally advanced disease at presentation. Reasonably good rates of good pathological response were observed. The work's strength are the translational correlates, many of which have not been previously described in this setting.

Response: Dear professor, thank you for positive comments and detailed suggestions, which have helped us improve the manuscript. We have read through all of your comments, responded to them point-by-point, and carefully revised the manuscript accordingly. We have made substantial amendments to the manuscript, mainly in four areas:

1. **Clinical:** We organized several meetings to discuss your comments with clinical doctors in medical oncology, gastrointestinal surgery, pathology, and radiology. We re-explained the results that were in your doubt, updated the follow-up of all patients to reach a median of 24 months, and collected and reported the post-operation complications.
2. **Statistics:** We discussed the statistical questions with the statistical methodology team at our institution several times. This team is led by Dr. Ming Lu. The team helped re-perform all of the statistical analyses and proof the text.
3. **Bioinformatics:** The bioinformatics work was revised substantially. Our team re-analyzed all of the data for microsatellite stable (MSS) patients separately, used three different mutation callers to recall the mutations for more reliable results, used a simulation method to balance tumor purities for tumor mutational burden comparison, and performed Immunophenoscore analysis for all paired specimens.
4. **Multiplex immunofluorescence:** We performed multiplex immunofluorescence for additional patients with non-MPR as a side comparison.

With regard to your comments and suggestions, we wish to reply as follows. For your convenience to review, we uploaded two manuscripts. One is with track changes, its line numbers being referred to by the response letter. A clean version was in the supplementary files, with the revisions colored in red.

Introduction: This is well written and highlights the specifically poor outcomes of the patients included in the current trial.

Response: Thanks for your affirmation and encouragement.

Results:

1. Suggest changing "median observation" to "median follow up"?

Response: Thank you for the suggestion. We have replaced the word "observation" with "follow-up". This change makes the results more accurate. (**Results, lines 86 - 87**)

2. CONSORT diagram – does this need to be separated into T4a vs T4b? It's a bit confusing. Suggest for the description in the paper to deal with all patients and have CONSORT reflecting this with supplementary CONSORT describing T4a and T4b?

Response: Thank you for this comment. We agree that the CONSORT diagram should be drawn as a single-arm as in the text (**Results, lines 82 - 84**), where we described the patients together. According to your suggestion, we modified the CONSORT diagram by drawing a general arm for all patients, with two branches describing T4a and T4b patients as supplementary (**Supplementary Fig. S1**). We believe the new CONSORT diagram will be clearer for the readers.

3. Did patients with peritoneal mets have laparoscope pre treatment?

Response: Thank you for this question. These patients did not receive laparoscope pre-treatment; they only received imaging examinations to exclude peritoneal metastasis. We agree that laparoscope before treatment is better for clinical staging, although it hasn't become routine in our center yet. We will consider

pre-designing laparoscope exploration before the therapy in future trials. For this point, we explained it in the manuscript (**Methods, line 427 - 428**).

4. Of interest, 4/5 of the MPR patients were already predisposed to have a response to anti-PD-1 – only one was MSS and PD-L1 negative. This is the patient of interest. For the others, I am not sure that we can comment that antiangiogenic therapy added any value.

Response: Thank you for this feedback. The MSI-H patients or PD-L1-high patients are considered beneficiaries of ICI. The two indicators partially explain the good pathological responses in these patients. However, a good response was not guaranteed in gastric cancer. Primary resistance of MSI-H tumors to immunotherapy has recently attracted wide attention and become a hot research issue. In advanced MSI-H gastric cancer the ORR rates for ICI monotherapy were only 47-57% in KEYNOTE-059, -061, and -062 trials (Chao, *J Clin Oncol*.2020.38.4_suppl.430). Even ICI plus chemotherapy in checkmate-649 obtained an ORR rate of only 55%, and about 30% of MSI-H patients were deceased within one year (Janjigian et al., 2021 ESMO, LBA 7). Furthermore, the PD-L1 ≥ 5 population showed a 60% ORR rate for ICI plus chemotherapy (Janjigian, *Lancet*, 2021). In the neoadjuvant setting in our trial, all MSI-H/PD-L1+ patients received MPR, which does not exclude the possibility that apatinib might add some efficacies in this trial. However, this is uncertain, given the small sample size.

In fact, due to the uncertain efficacy of ICI monotherapy in gastric cancer in 2019, especially lacking any outcome in the neoadjuvant setting, we had to design this combined treatment to expect a more reliable efficacy and to meet the ethical requirements to protect patients. Apatinib was found to synergize with camrelizumab in preclinical studies (Zhao, *Cancer Immunol Res*, 2019) and has a 16% objective response rate in refractory advanced GC (Xu, *Clin Cancer Res*, 2019). It was shown to alleviate hypoxia, increase infiltration of CD8⁺ T cells, reduce recruitment of tumor-associated macrophages in tumors and decrease TGF β in mouse models (Zhao, *Cancer Immunol Res*, 2019). The synergy of apatinib and camrelizumab was also observed in hepatocellular cancer (Xu, *Clin Cancer Res*, 2021), SCLC (Fan, *J Thorac Oncol*, 2021), and cervical cancer (Lan, *J Clin Oncol*, 2020)

As you noted, the single-arm design with combined treatment could not distinguish the relative contributions of each component in either MSI-H or MSS patients. A side comparison with control groups is needed for this purpose. These concerns have been discussed in the revised manuscript (**Discussion, lines 401 - 403; Introduction, line 68 - 70**), and we expect to address these issues in large controlled trials in the future.

5. Post-op complications need to be reported in much more detail, especially in view of the close relationship between antiangiogenic therapy and surgery. Please report according to Clavien Dindo and comment on ST03 and RAMSES trial results

Response: Thank you for providing this suggestion. We sincerely agree that postoperative complications are essential for trials with neoadjuvant therapy. As you have suggested, we analyzed and reported the postoperative complications in the revised manuscript. There were no high-morbidity complications; all were grade 1 or 2 according to the Clavien-Dindo classification (**Methods, lines 432 - 433**). We added a paragraph (**Results, lines 152 - 156**) and a supplementary figure (**Supplementary Fig. S4**) to describe these complications in the manuscript.

We also added comments on ST03 and RAMSES trial results and compared them with ours (**Discussion, lines 329 - 338**). In these two trials, bevacizumab and ramucirumab increased surgical morbidity rates when combined with neoadjuvant chemotherapy, especially the anastomotic leakage and wound healing complications (Cunningham, *Lancet Oncol*, 2017; Al-Batran, ASCO, 2020). By contrast, the treatment with apatinib in our preliminary results didn't have high-incidence morbidities, and there was only 1 (5%) anastomotic leakage and 1 (5%) wound healing complication. This is consistent with another trial with apatinib for conversion therapy of gastric cancer (Xu, *Front Pharmacol*, 2021). The underlying reason might be the much shorter half-life of apatinib compared to antibodies (9 hours vs. 10-20 days; Roviello, *Cancer Letters*, 2016). The 14-day interval in our design should be long enough to be safe for surgery.

6. Is it possible to report the TMB and TNB results separately for MSS vs MSI patients? It's described already that TMB in MSI patients will decrease with anti-PD-1 (Kwon et al, Cancer

Discovery 2021). It's not been done for MSS, and I suspect the result overall is driven by the MSI results.

Response: Thank you for providing this valuable suggestion, which we think is essential. We re-performed the TMB/TNB analyses by excluding MSI-H patients and found the TMB was still significantly decreased in the MSS population (**Fig. 3D**). We also used a method to prove that the decreased TMB is not just due to the drop in purity (**Fig. 3E**). These data were explained in the manuscript (**Results, lines 163 - 165**).

7. I am not sure that the sample size is powered adequately for "gain" and "loss" analysis as described so would not place too much emphasis on this.

Response: Thank you for your suggestion. We agree that the 15 pairs of samples are not enough to have adequate power. Therefore, we softened the description of these analyses by deleting the detailed description of specific patients, leaving only the correlation result (**Results, lines 190 - 194**). We also discussed the small size as a limitation in the Discussion section (**Discussion, lines 394 - 397**).

8. It's not surprising that immune inflamed genes were upregulated in responders as most of these were MSI and/or Pd-L1 positive.

Response: Thank you for this comment. We agree that the MSI-H and PD-L1 positive tumors commonly have high levels of inflamed genes. In response to your comment, we excluded the MSI-H/PD-L1 positive samples and repeated the analysis (**Supplementary Fig. S9B; Results, lines 228 - 229**). The MSS PPR+ sample have similar levels of immune-related genes to the MSI-H/PD-L1 positive patients in most analyses. Unfortunately, statistical tests cannot be performed because only 1 MSS PPR+ sample was available. We also mentioned that the high levels of immune inflamed genes and immune infiltrations were partially due to MSI-H patients (**Discussion, lines 355 - 356**).

9. The changes in the TME in responding patients have also been demonstrated in patients responding to chemotherapy (Lee et al, *Cancer Discovery* 2021). Can the authors comment on any added value that might have been gained from apatinib?

Response: Thank you for this question. We noticed that Lee et al. reported the early tumor-immune microenvironmental remodeling and responses to first-line chemotherapy in advanced gastric cancer in small sizes (12 pre-treatment patients and 6 on-treatment patients). In their study, responders showed increases in NK cells, M1 macrophages, and effector T cells and decreases in M2 cells, while the non-responders are characterized by increased B cells and LAG3-expressing T cells. The immune activation might result from chemotherapy-induced immunogenic cell death (ICD). Our analysis found partially similar changes, including increased M1 and CD8⁺ T cells in the responders. Besides, we also found decreased T_{reg} cells and increased DC cells, which might be contributed by apatinib or ICI or their synergic effects with chemotherapy. Yet, this comparison and deduction were not solid.

Apatinib was also shown to remodel TME by increasing infiltration of CD8⁺ T cells and suppressing tumor-associated macrophages in animal models (Zhao, *Cancer Immunol Res*, 2019). Combined with ICI, it showed excellent efficacy in advanced gastric cancer (Xu, *Clin Cancer Res*, 2019) and other cancer types (Xu, *Clin Cancer Res*, 2021; Fan, *J Thorac Oncol*, 2021; Lan, *J Clin Oncol*, 2020). Based on these, we designed this combined treatment for more reliable efficacy. However, as mentioned above, the individual contribution of apatinib was hard to determine, considering this study's single-arm design. A side comparison with control groups is needed for this purpose.

The above comments have been discussed in the revised manuscript (**Discussion, lines 375 - 378; Discussion, lines 401 – 403; Introduction, lines 68 - 70**). We think the *Cancer Discovery* paper is important and also discussed it in the manuscript (**Discussion, line 375 - 378**). We agree with you that determining the added value of apatinib is an essential issue, and we look forward to addressing this issue in large controlled trials in the future.

Discussion:

- 1. The statement that neoadjuvant ICI is superior to chemotherapy in pCR is overstated. Path CR was 25% and this is not much more than is reported with FLOT. Also the sample size is very small.**

Response: Thanks for this feedback. Indeed, the FLOT4-AIO trial showed a numerically similar pCR rate, and we should not overstate it.

It is worth noting that the patients in our cohort (all cT4N+ and 56% cT4bN+) are more advanced than those in the FLOT4 trial (cT4 patients < 10%), being predisposed to a worse prognosis. In addition, in FLOT4, pCR mainly occurs in the intestinal type, with only 3% in the diffused type (Al-Batran, *Lancet Oncol*, 2016). By contrast, all pCR patients were diffused types in our study, and their pCR rate was 30% (3/10). These differences were briefly discussed in the manuscript (**Discussion, lines 303 - 306**). As you suggested, we have softened the statement by deleting the word "superior" in the revised manuscript (**Discussion, lines 295 - 296**). We also read through the entire manuscript and softened similar statements (e.g., **Conclusion lines 408 - 411**). We believe these changes make the discussion more accurate.

- 2. The statement that 293 genes were detected that might predict response is not valuable. Even by multiplicity some genes would be likely to be positive. This number of genes is not likely to be useful.**

Response: Thank you so much for this advice, which another reviewer also pointed out. Based on comprehensive discussions, we agree that the 293 genes may not be valuable in predicting responses, as you mentioned. Therefore, we deleted the results of the 293 genes from the manuscript (**Results, lines 204 - 208 and Discussion, 352 - 353**).

- 3. There is not a lot of discussion around what the apatinib added, if anything. This should be mentioned in more detail.**

Response: Thank you for this suggestion. Considering the uncertain efficacy of ICI monotherapy in gastric cancer in 2019, especially in the neoadjuvant setting risking surgery delay and inoperativeness, we designed this combined treatment to expect a more reliable efficacy and to meet the ethical requirements. Apatinib had been approved for the 3rd-line treatment of gastric cancer in China, and it, together with camrelizumab, reached a 16% objective response rate in refractory GC (Xu, *Clin Cancer Res*, 2019). Therefore, we adopted apatinib in this single-arm study.

We have adjusted the manuscript accordingly. As mentioned above, the previous publications (Zhao, *Cancer Immunol Res*, 2019) showed apatinib enhanced anti-tumor cell infiltration, so we infer that it may also synergize with ICI in this trial (**Introduction, lines 68 - 71**). Through comparison with the *Cancer Discovery* paper you recommended, we found our treatment specifically enhanced DC cells in responders (Lee et al., *Cancer Discovery*, 2021) (**Discussion, lines 375 - 378**). However, the single-arm design makes it challenging to dissect the relative contributions of apatinib. We have mentioned this limitation in the revised manuscript (**Discussion, lines 401 - 403**). We expect to address this issue in controlled trials in the future.

4. Overall, the conclusions need to be tempered by the fact that a large proportion of responding patients in the trial had MSI tumours and PD-L1 negative tumours (certainly MSI tumours were more than the 6-8% expected...).

Response: Thank you for this valuable suggestion. The percentage of MSI-H patients in our study was about 16%. In this study, the patients were recruited in chronological order, with no preference for any pathological type. The high percentage of MSI-H patients may be due to the small sample size. We agree with you that the conclusion should be tempered to be more precise. We have revised the conclusions by emphasizing that the MSI-H or PD-L1 positive patients account for many responders and how to improve its efficacy in MSS and PD-L1 negative patients need further exploration (**Conclusion, lines 408 - 411**). We believe these changes make the discussion and conclusion more accurate.

In addition, as we mentioned above, MSI-H patients only have about 47–57% ORR rates in first-line treatment, and, to our knowledge, there are no published articles reporting the pathological regression in MSI-H gastric cancer after ICI or combination therapy in the neoadjuvant setting. In this sense, our data on MSI-H/PD-L1+ gastric cancer patients with 100% MPR may be interesting.

Again, we thank you for carefully reviewing our manuscript and offering constructive suggestions, which have greatly improved the study.

Reviewer #4 (Remarks to the Author): with expertise in biostatistics, clinical trial study design

The statistical analyses have many awkward and incorrect descriptions, and analyses themselves have many flaws. The following is the list of the problems, all of which must be addressed for this report to be considered further. The statistical analysis plan must be completely rewritten with enough details so that the readers know what methods were used. It is essential that a qualified statistician handle the statistical analysis.

Response: Dear professor, thank you for your valuable and detailed suggestions, which have helped us improve the manuscript. We realized our lack of experience in statistics. As you suggested, we sought assistance from experienced statisticians during revision. We discussed all statistics with the statistical methodology team led by Dr. Ming Lu from our institution several times. A professional statistician reperformed all of the statistical analyses, proofed the texts, and helped to re-write the statistical analysis plan. We have read through all of your comments, responded to them point-by-point, and carefully revised the manuscript accordingly.

With regard to your comments and suggestions, we wish to reply as follows. For your convenience to review, we uploaded two manuscripts. One is with track changes, its line numbers being referred to by the response letter. A clean version was in the supplementary files, with the revisions colored in red.

1. Please include a sample size justification.

Response: We appreciate your valuable comments on this manuscript. We believe that a sample size calculated based on an assumption is the best for this study. However, it was challenging to make an assumption for pathological response rates, which is our primary research endpoint, when we designed this study in 2019. At that time, ICI showed some efficacy (only radiological response or survival outcome) in some studies in the late-line setting, such as in the ATTRACTION-02 trial (Kang, *Lancet*, 2017) and KEYNOTE-059 trial (Fuchs, *JAMA Oncol*, 2018). In the second-line setting, ICI failed to prolong survival (e.g., KEYNOTE-061; Shitara, *Lancet*, 2018). Plus, the trial outcomes in the first-line setting were not

reached (like Checkmate-649 and KEYNOTE-062). Most importantly, there were no reported trials on neoadjuvant ICI-based treatment in gastric cancer, so we were unable to estimate a pathological response rate for sample size calculation. Therefore, we set a sample size for an exploratory study following similar trials in lung cancer and melanoma (Forde, *NEJM*, 2018 and Amaria, *Nature Medicine*, 2018). In addition, recent similar studies have all recruited about 20–30 subjects (Chalabi, *Nature Medicine*, 2020 and ASCO meeting abstracts DOI: *J Clin Oncol*.2021.39.15_suppl.4040, 4046, and 4026). Actually, it was not until 2021 that some neoadjuvant ICI studies in gastric cancer reported their pathological response rates in meetings (e.g., 2021 ASCO abstracts DOI: *J Clin Oncol*.2021.39.15_suppl.4040, 4046, and 4026). We have summarized these abstracts as a meta-analysis (2022 ASCO-GI abstract DOI: *J Clin Oncol*.2022.40.4_suppl.291). These results will help us determine sample size in our future study. We will use the available outcomes to calculate sample sizes with the help of statisticians in our future trials.

2. Line 402: "Kaplan-Meier" is a name of an estimator. There is no "Kaplan-Meier analysis".

Response: Thank you for pointing out this incorrect description. After reading your comment and related literature, we realize that it should be "survival analysis" or "Kaplan-Meier estimates of survival" (Statistics review 12, survival analysis. *Crit Care*. 2004). We have made a change in the revised manuscript (**Methods, lines 487 – 489, and Methods, lines 575 - 576**).

3. Line 470: Comparison is not estimated.

Response: Thank you for your comment. We have changed the wording to "... test was used for comparison..." in the revised manuscript (**Methods, lines 566 - 570**).

4. Line 471: Wilcoxon signed-rank test / pair-wise Wilcoxon signed-rank test are incorrect tests. Wilcoxon rank sum test should be used for comparison of independent groups.

Response: Thank you for sharing your expertise. We apologize for misusing the name of these statistical tests. After confirming with statisticians, we learned that the names should be "Wilcoxon Rank-Sum test"

(for independent groups) and “Wilcoxon Signed-Rank test” (for paired samples). A statistical expert carefully proofed all the statistical analyses by re-calculating the significances using the proper tests (**Methods, lines, 566 - 570**). We believe that accurate descriptions of statistical tests are essential for any manuscript, and your comments are valuable not only to this manuscript but also to our future work.

5. Line 472: "... regression were tested" is unclear.

Response: Thank you for your feedback. We have changed the wording to, “Correlations between biomarkers and pathological regressions were assessed by Spearman’s rank correlation coefficient” (**Methods, lines 573 - 575**).

6. Line 473: "Pearson" is a name of an estimator (estimate), and it is not a name of a test.

Response: Thank you for this comment. We have revised this sentence to, “Correlations ... were assessed by Spearman’s rank correlation coefficient.” (**Methods, lines 573 - 575**).

7. Line 473: There is no "Log-rank regression".

Response: Thank you for the correction. We learned that it should be “log-rank test,” and we have changed the revised manuscript (**Methods, lines 575**).

8. Line 426: Plots are not conducted.

Response: Thank you for this correction. We have revised the sentence to, "Plots are drawn...." (**Methods, line 521**).

9. Throughout the manuscript, an estimate of the median is followed by a 95% confidence interval. Please describe/explain how this confidence interval is calculated. It is more common to use lower and upper quartiles to indicate data variability.

Response: Thank you for this important suggestion. The median with a 95% confidence interval was calculated by the one-sample Wilcoxon signed-rank test (<https://datatricks.co.uk/one-sample-wilcoxon-test-in-r#:~:text=A%20one-sample%20Wilcoxon%20test%20is%20used%20to%20test,assumed%20that%20the%20data%20are%20not%20normally%20distributed>). However, after reading your feedback and consulting with statisticians, we realized that these data distributions should be described by medians and interquartile ranges, instead of 95% confidence intervals. Therefore, we made replaced 95% CI with IQR in the revised manuscript (**Lines 87, 95, 101, 109, 149, and 155**).

10. Line 425: "Log2FoldChange > 1.5" is strange. Does this mean FoldChange > 2^{1.5}? If the authors really mean "Log2FoldChange > 1.5", this cut off corresponds to 2.82 fold changes and seems unjustified.

Response: Thank you for pointing out this problem. We re-checked the manuscript and figures and found they were described incorrectly. The threshold for the differential expression gene is $|\log_2\text{FoldChange}| > 2$. We have corrected them in the manuscript (**Methods, line 520**).

Figures:

1. Boxplots are used to show the distributions, but given small sample sizes, boxplots are not optimal ways to show data. As the authors did in some places, dots and lines connected pre- and post- observations are effective ways to show these data. (e.g., Figure S8a). Remove boxes and just show dots and lines. Additionally, dots showing differences (post - per) may be informative.

Response: Thank you for this comment. We agree that the box plots are not optimal. We have re-drawn all of the plots (**Fig. 3B-E, 4D-E, 4G-H, 5B-D, 6E, S5B, S9-13, and S16**). We used dots and lines to show paired groups with pre- and post-treatment data. For comparison between independent groups, we used dots plus boxes.

2. Figure S4 and its associated regression analysis are highly problematic. Because of a few outliers and influential points and abundance of zeroes, most of the fitted lines do not represent the data well. Some fitted lines go below zero. The whole analyses associated with this figure must be redone with appropriate considerations given to outliers, influential data, and zeroes. Removing them would be inappropriate.

Response: Thanks for your comment. We agree that the regression analysis is problematic. This figure was used to show the distribution of gene VAF during PyClone construction instead of to study their correlation. Therefore, the regression analysis was unnecessary, and the regression lines were removed from this figure.

3. Line 419 and other places: The word, "data", is plural.

Response: Thank you for pointing out this incorrect grammar. We have corrected this mistake in the revised manuscript ("...data from the sequencer were processed ..."; **Methods, line 515**).

We thank you again for carefully reviewing our manuscript and for giving us critical and constructive suggestions that have greatly improved our manuscript and our future study.

REVIEWERS' COMMENTS

Reviewer #1 (Remarks to the Author):

In their revised version the authors have undertaken considerable effort to address the reviewer concerns. The additional analyses are informative and the revised version is an improve manuscript. Although the findings should all be considered exploratory there is biologic rationale and directionality to the data. I would consider clarifying the following points

1. Would note specifically where analyses we done post-hoc. For example, the TRG assessment appears to have been done in response to reviewer comments and not originally planned and therefore should be called out as post-hoc. This applies to other analyses not originally planned.
2. Given the availability of RNA-seq data it would be of interest to analyze the TME subtype and include this in the figures. This would add a validated metric to the paper and would be of interest. Should be readily doable. Reference is, Bagaev A, Kotlov N, Nomie K, et al. Conserved pan-cancer microenvironment subtypes predict response to immunotherapy. Cancer Cell. 2021 Jun 14;39(6):845-865.e7. PMID: 34019806.
3. The SSPO and RREB1 associations with MPR remain hard to interpret and to biologically understand. I would consider emphasizing these point unless there is additional mechanistic data.

Reviewer #2 (Remarks to the Author):

The authors have addressed most of my main concerns. My key remaining major concern relates to the reported mutations in SSPO. From the authors response it is now apparent that this gene is a non-translated pseudo-gene.

1)

The authors must to be careful when they talk about silent and non-silent (splice-site?) mutations in this pseudogene:

E.g. "Patients with SSPO mutation, including both silent and non-silent mutations, ". Please add context (mention that SSPO is a pseudogene) and fix this sentence.

2)

The mutations in this gene (and another new candidate gene) are prominently highlighted in the abstract: "examination revealed novel biomarkers for pathological responses, including RREB1 and SSPO mutation, ". Firstly, all the reported novel correlations in this study have not been validated in independent cohorts, they are therefore 'candidate' or 'putative' biomarkers. Please rephrase. Secondly, the authors provide no context on the potential function of the mutations in SSPO. I would encourage the authors, for full transparency, to highlight in the abstract and on p. 8 (first time SSPO is mentioned) that SSPO is a pseudogene, and that the function of the mutations are unknown.

Reviewer #3 (Remarks to the Author):

Congratulations to the authors for their comprehensive response.

Reviewer #4 (Remarks to the Author):

The authors have addressed every question and comment that I had on the original submission. Three minor points to note:

1. IQR is the difference of upper quartile and lower quartile, thus, it is a single number. Please replace 'IQR' by 'Quartiles' when you want to list both numbers (lower quartile, upper quartile).

2. Regarding the original issue #10, the new " $\log_2\text{foldchange} > 2$ " indicates that you require fold change > 4 or $< 1/4$. Please confirm that this is correct. It is better to simply state this in the original scale without taking logs, i.e., Fold change > 4 (or perhaps Fold change > 2).

3. Please pay attention to the significance digits of p-values. " $P=0.5$ " (too few digits) and " $P=0.9034$ " (too many digits) are both inappropriate.

Reviewer #1 (Remarks to the Author):

In their revised version the authors have undertaken considerable effort to address the reviewer concerns. The additional analyses are informative and the revised version is an improve manuscript. Although the findings should all be considered exploratory there is biologic rationale and directionality to the data. I would consider clarifying the following points:

Response: Dear professor, we appreciate your valuable comments, which have driven the improvement of this manuscript. We also thank you for your further advice. We would like to respond as follows:

1. Would note specifically where analyses we done post-hoc. For example, the TRG assessment appears to have been done in response to reviewer comments and not originally planned and therefore should be called out as post-hoc. This applies to other analyses not originally planned.

Response: Thank you for your suggestion, which we believe is important. In this study, the Becker TRG assessments, mlHC, Immunophenoscore, and TME subtyping were performed post-hoc. To revise, we highlighted that they were analyzed post-hoc in the **Methods (Lines 411-412)**.

2. Given the availability of RNA-seq data it would be of interest to analyze the TME subtype and include this in the figures. This would add a validated metric to the paper and would be of interest. Should be readily doable. Reference is, Bagaev A, Kotlov N, Nomie K, et al. Conserved pan-cancer microenvironment subtypes predict response to immunotherapy. Cancer Cell. 2021 Jun 14;39(6):845-865.e7. PMID: 34019806.

Response: This advice is quite valuable. The study you referred provides a practical subtyping method for solid tumors in the setting of immunotherapy. According to this study, we constructed a KNN model trained by TCGA STAD samples using the CLASS package. Then, our data were classified by this KNN model. The pre-treatment tumors were rarely (2/12) "Immune-enriched". After the treatment, many tumors, including all PPR+ and 5/9 non-PPR+ tumors, shifted from "depleted" or "fibrotic" to "immune" or "immune/fibrotic" types. This information was added in **Fig. 5A and 5E, Figure legends (lines 1040-1042)**, and **Results (lines 231-233)**. The KNN method was also described in the **Methods (lines 506-510)**.

3. The SSPO and RREB1 associations with MPR remain hard to interpret and to biologically understand. I would consider emphasizing these point unless there is additional mechanistic data.

Response: Thank you for this suggestion. We agree with you that the underlying mechanisms of this association between *SSPO/RREB1* mutations and responses were unknown, so we highlighted this in the **Abstract (lines 39-40)** and **Discussion (lines 323-325)**. Besides, the correlations between *SSPO/RREB1* mutations and responses were not validated in other independent cohorts, so we also emphasized that they were only "putative biomarkers" in the manuscript (**Abstract, lines 37; Results, line 174; Discussion, lines 328-329**).

We thank you again for carefully reviewing our manuscript and offering critical and constructive suggestions, which significantly improve our manuscript.

Reviewer #2 (Remarks to the Author):

The authors have addressed most of my main concerns. My key remaining major concern relates to the reported mutations in SSPO. From the authors response it is now apparent that this gene is a non-translated pseudogene.

Response: We appreciate your valuable comments, which have greatly improved this manuscript. We also thank you for your further advice related to SSPO. We would like to respond as follows:

1) The authors must to be careful when they talk about silent and non-silent (splice-site?) mutations in this pseudogene: E.g. "Patients with SSPO mutation, including both silent and non-silent mutations,". Please add context (mention that SSPO is a pseudogene) and fix this sentence.

Response: We agree with you that talking about 'silent' and 'non-silent' mutations on a pseudogene is inappropriate. According to your suggestion, we mentioned that "SSPO is a pseudogene" before this sentence (**Results, line 174**) and deleted the description of "silent and non-silent mutations" (**Results, line 175**). Besides, we checked throughout the manuscript and found that "SSPO non-silent mutation" was also used elsewhere (**Results, lines 176-177**), so we deleted it. We believe these revisions make the description clearer.

2) The mutations in this gene (and another new candidate gene) are prominently highlighted in the abstract: "examination revealed novel biomarkers for pathological responses, including RREB1 and SSPO mutation,". Firstly, all the reported novel correlations in this study have not been validated in independent cohorts, they are therefore 'candidate' or 'putative' biomarkers. Please rephrase. Secondly, the authors provide no context on the potential function of the mutations in SSPO. I would encourage the authors, for full transparency, to highlight in the abstract and on p. 8 (first time SSPO is mentioned) that SSPO is a pseudogene, and that the function of the mutations are unknown.

Response: Thank you for pointing out this issue. We agree that the putativity of the biomarkers and the functions of SSPO should be highlighted clearly for transparency. According to your suggestion, we highlighted that the 'biomarkers' were putative in this study in the **Abstract (line 37)** and **Results (line 174)**. We mentioned that SSPO is a pseudogene in humans with unknown functions in cancer in the **Abstract (lines 39-40), Results (line 174), and Discussion (lines 323-325)**. Besides, we emphasized that the association between *SSPO/RREB1* mutations and responses was not validated in other independent cohorts in the **Discussion (line 328-329)**.

Again, we thank you for carefully reviewing our manuscript and providing critical and constructive suggestions, which have greatly improved our manuscript.

Reviewer #3 (Remarks to the Author):

Congratulations to the authors for their comprehensive response.

Response: Dear professor, we appreciate your help, which has dramatically improved this manuscript.

Reviewer #4 (Remarks to the Author):

The authors have addressed every question and comment that I had on the original submission. Three minor points to note:

Response: We appreciate your valuable comments, which have driven the improvement of this manuscript. We also thank you for your further advice. We would like to respond as follows:

1. IQR is the difference of upper quartile and lower quartile, thus, it is a single number. Please replace 'IQR' by 'Quartiles' when you want to list both numbers (lower quartile, upper quartile).

Response: Thank you for this suggestion. We have changed 'IQR' to 'Quartiles' in the manuscript **(Results, lines 91, 97, 103, 111, 148, 155, and 547)**

2. Regarding the original issue #10, the new " $\log_2\text{foldchange} > 2$ " indicates that you require fold change > 4 or $< 1/4$. Please confirm that this is correct. It is better to simply state this in the original scale without taking logs, i.e., Fold change > 4 (or perhaps Fold change > 2).

Response: We appreciate this comment. We rechecked the data and confirmed that it indicated fold change > 4 or $< 1/4$. According to your comment, we have changed " $|\log_2\text{foldchange}| > 2$ " to "fold changes > 4 or $< 1/4$ " **(Methods, line 493)**.

3. Please pay attention to the significance digits of p-values. "P=0.5" (too few digits) and "P=0.9034" (too many digits) are both inappropriate.

Response: Thank you for pointing out this issue. We have checked the manuscript, figures, and supplementary figures. We kept two significant digits when p-value > 0.0010 and one significant digit when p-values < 0.0010 **(Figs. 1D, 2C, 2E, 2F, 4G, 5C, 6C, 6E, S5B, S9, S10, S11, S12, S13, and S16)**.

Again, we thank you very much for carefully reviewing our manuscript and providing critical and constructive suggestions, which have greatly improved our manuscript.